# Selling Data as a Digital Good with Scaling Valuations

**Ningyuan Li** [* 1]  **Yanru Guan** [* 2]  **Xiaotie Deng** [1]  **Zihe Wang** [3]  **Jie Zhang** [4]

## Abstract

We study mechanism design for selling data as a digital good when the value derived from training AI models follows a scaling law. The seller faces a linear cost when producing data, while the buyers benefit from additional data with diminishing returns as data volume increases. This departs from classical auction models by allowing allocations to be continuous quantities of data rather than binary outcomes. We first analyze an offline setting in which all buyer types are realized simultaneously, characterizing profit-optimal mechanisms and showing how virtual-value methods extend to continuous data allocations. We then consider an online setting with sequential arrivals, where production decisions must be made under demand uncertainty. We show that myopic allocation and fixed production plans can be arbitrarily suboptimal, whereas a simple two-stage algorithm that combines upfront production with adaptive expansion achieves a constant-factor approximation to the offline optimum. Finally, we study bilateral data trading under asymmetric information, where both the buyer's value and the seller's cost are private. Although the optimal truthful mechanism has a complex structure, we show that simple and implementable mechanisms recover a constant fraction of the *first-best gain-from-trade*. Overall, our results highlight how scaling laws introduce new algorithmic trade-offs in market design and provide performance guarantees for data markets under uncertainty.

---

[*]Equal contribution  [1]Center on Frontiers of Computing Studies, School of Computer Science, Peking University, Beijing, China [2]School of Electronics Engineering and Computer Science, Peking University, Beijing, China [3]Gaoling School of Artificial Intelligence, Renmin University of China, Beijing, China [4]Department of Computer Science, University of Bath, Bath, United Kingdom. Correspondence to: Xiaotie Deng <xiaotie@pku.edu.cn>, Zihe Wang <wang.zihe@ruc.edu.cn>, Jie Zhang <jz2558@bath.ac.uk>.

*Proceedings of the 43$^{rd}$ International Conference on Machine Learning*, Seoul, South Korea. PMLR 306, 2026. Copyright 2026 by the author(s).

## 1. Introduction

Large-scale datasets have become a central input to modern artificial intelligence systems. Across a wide range of model classes, empirical evidence shows that model performance improves with data volume according to scaling laws, with diminishing returns as the amount of training data increases (Kaplan et al., 2020; Iansiti, 2021; Rekatsinas et al., 2014; Zheng et al., 2017; Amsterdamer et al., 2011). These observations suggest that data has a structured but nonlinear value, raising fundamental questions for market design. How should a seller decide how much data to produce, and how should prices and allocations depend on buyer valuations when the value derived from training AI models scales sublinearly with data volume?

A key economic distinction relative to classical auction markets is that data is non-rivalrous. Once collected, the same dataset can be copied and shared with multiple buyers without depletion. As a result, the primary constraint faced by a seller is the cost of producing data, rather than exclusivity in allocation. Allocating more data to one buyer does not reduce what can be allocated to others. Instead, the central trade-off is between expanding data production and the total value generated across buyers.

Motivated by these features, we propose a mechanism design framework for selling data as a digital good. Buyer valuations are modeled to scale as a power law in data volume, capturing the empirical scaling effects observed in machine learning. This implies that marginal value declines as more data is allocated, leading to nonlinear and state-dependent allocation rules. At the same time, the seller incurs explicit production costs and must decide how much data to generate in response to demand. Together, diminishing returns in buyer value and costly data production fundamentally distinguish this setting from classical auction models and require new analytical tools.

We first study an offline setting with multiple buyers, where all private information is realized simultaneously. In this environment, we characterize profit-optimal mechanisms and show how virtual-value-based methods extend to continuous data allocations under scaling effects. Unlike classical auctions, optimal mechanisms may allocate the same dataset to multiple buyers and pool demand to justify data production.

We then turn to an online setting in which buyers arrive sequentially and the seller must make irreversible production decisions under uncertainty. In this environment, the tight connection between production and allocation no longer holds. We show that myopic allocation rules and fixed, non-adaptive production plans can perform arbitrarily poorly. In contrast, a simple mechanism that combines initial data production based on prior distributions with adaptive expansion over time achieves a constant-factor approximation to the offline optimum.

Finally, we study bilateral data trading under asymmetric information, motivated by settings such as regulated data sharing or government-mediated data markets. In this model, both the buyer's value and the seller's cost are private, and the mechanism is designed by a third party to maximize *gain-from-trade.* We characterize the optimal truthful mechanism in this setting and show that simpler, implementable mechanisms can still recover a constant fraction of the *first-best welfare*, with guarantees that depend explicitly on the scaling parameter.

Taken together, our results clarify how non-rivalry, scaling laws, and production costs reshape classical mechanism design insights for profit and gain-from-trade optimization, and give rise to new algorithmic trade-offs in online and bilateral data markets.

## 1.1. Related Work

**Data markets and selling information.** Admati & Pfleiderer (1986) study the problem of selling information using a monopolistic model. Subsequently, data markets are studied as distinct economic institutions by Balazinska et al. (2011). Along this line, Koutris et al. (2015) develop query-based data pricing models, and Agarwal et al. (2019) study two-sided data marketplaces. In practice, a variety of data marketplaces have emerged, including AWS Data Exchange (Amazon Web Services, 2024) and Snowflake's data marketplace (Snowflake, 2024), illustrating the growing commercialization of data. For a comprehensive overview of data markets, we refer the reader to the survey by (Zhang et al., 2024).

**Mechanism design for data markets.** More recently, data markets have attracted growing attention from a mechanism design perspective. In this line of work, the designer's objective is typically either welfare maximization, as in (Grubenmann et al., 2018; Wang et al., 2016; Cao et al., 2017; Rasouli & Jordan, 2021), or revenue maximization, as in (Agarwal et al., 2019; 2024; Bonatti et al., 2024). Notably, Agarwal et al. (2019) depart from much of the earlier literature by modeling buyers as purchasing improvements in predictive performance rather than raw data itself, while Agarwal et al. (2024) and Bonatti et al. (2024) incorporate

externalities arising from data usage and sharing.

Most of these works implicitly assume that data is already collected and treated as a fixed, well-defined commodity, so data production is not a decision variable for the seller. In many real-world settings, however, data does not exist ex ante and must be deliberately produced in response to buyers' needs. For example, data sellers on platforms such as Datatang collect datasets through crowdsourcing, including images of sign language gestures and 3D hand gestures (Datatang, 2024). Moreover, the quantity of data directly affects buyers' utility. Motivated by both empirical and theoretical evidence, we therefore treat data production as an explicit decision variable and assume that the value of data exhibits diminishing marginal returns as data quantity increases (Kaplan et al., 2020; Iansiti, 2021; Rekatsinas et al., 2014; Zheng et al., 2017; Amsterdamer et al., 2011).

**Combinatorial auctions.** Our model can be mapped to a combinatorial auction setting, as studied in (Dobzinski & Nisan, 2007a; Dobzinski, 2011; Deng et al., 2025). Each unit of data can be interpreted as an item, where all items are identical, and the value of a bundle depends only on its cardinality. Under diminishing marginal values, the induced valuation function satisfies the standard notion of submodularity used in the combinatorial auction literature. Dobzinski & Nisan (2007b) and Dobzinski et al. (2022) study multi-unit auctions, allowing buyers to be either unit-demand or multi-demand. In contrast, buyers in our model are not unit-demand, data can be sold repeatedly to multiple buyers, and, crucially, the seller's data quantity is an endogenous decision rather than a fixed supply. In the literature on digital goods, Goldberg & Hartline (2001), Hartline & Roughgarden (2008), and Chen et al. (2014) typically assume unlimited supply but restrict buyers to be unit-demand, which differs fundamentally from our setting.

**Online mechanisms and approximation guarantees.** Our results in Section 3 concern online data sales. Related online allocation problems are often analyzed using prophet inequalities, which yield constant-factor approximations via posted-price mechanisms computed from prior distributions (Krengel & Sucheston, 1977; Correa et al., 2019). These results rely on fixed supply or capacity constraints. In contrast, in our setting the supply of data is endogenous, and fixed production decisions based on the prior alone do not yield any approximation guarantee. We show that dynamic adjustment of data production in response to realized demand is necessary to achieve constant-factor performance.

**Bilateral trade and gain-from-trade.** There is a substantial literature on approximating gains from trade in bilateral settings with one buyer and one seller (Brustle et al., 2017; Deng et al., 2022). These models focus on indivisible goods

and do not consider quantity-dependent valuations. Our bilateral model in Section 4 differs by allowing continuous allocations governed by a scaling law and by leveraging a multiplicative welfare decomposition that leads to constant-factor approximations under simple truthful mechanisms.

## 2. Profit-Optimal Offline Mechanisms

In this section, we study the seller's optimal mechanism in an offline setting where all buyer types are realized simultaneously. We consider $n$ buyers with independent private types $v_1, \ldots, v_n$, where $v_i \sim F_i$ with density $f_i$. The seller can produce any amount of data at constant marginal cost $c > 0$. Producing a dataset of size $D \geq 0$ incurs total cost $cD$. Buyers first report their types, and then the seller determines the allocation and price. Unlike classical auction models with binary allocation, the seller here decides the amount of data allocated to each buyer. Data is non-rivalrous: once produced, the same dataset can be allocated to multiple buyers. The seller's objective is profit maximization rather than pure revenue maximization. We will provide a discussion on the modeling choices in Section 5.1.

Each buyer $i$ receives a *continuous* allocation $x_i \geq 0$ as the amount of allocated data, with a scaling valuation in a power-law form:

$$V_i(v_i, x_i) = v_i\, x_i^{\alpha}, \qquad \alpha \in (0, 1),$$

where $\alpha$ is an elasticity parameter capturing diminishing returns to data volume, consistent with empirical scaling laws in machine learning. If $\alpha = 1$, the model becomes the standard quasi-linear utility framework. We assume $\alpha$ is a publicly known constant.

The seller designs a direct-revelation mechanism $(x(\cdot), p(\cdot))$ with allocation rules $x(\cdot) = (x_1(\cdot), \ldots, x_n(\cdot))$ and payments $p(\cdot) = (p_1(\cdot), \ldots, p_n(\cdot))$ taking reported types as input. Given reported types $\hat{v}_1, \cdots, \hat{v}_n$, the utility of buyer $i$ with true type $v_i$ is

$$u_i(v_i, \hat{v}_i, \hat{v}_{-i}) = v_i\, x_i(\hat{v}_i, \hat{v}_{-i})^{\alpha} - p_i(\hat{v}_i, \hat{v}_{-i}).$$

If the seller produces a dataset of size $D$, a feasible allocation must satisfy $x_i \leq D$ for each buyer $i$, and producing more data than the largest allocation is never optimal. As a result, in any optimal mechanism the production level satisfies $D = \max_i x_i$.

**Incentive compatibility and individual rationality.** To incentivize truthful reporting of private types, the mechanism is constrained by Bayesian incentive compatibility (BIC) and individual rationality (IR). Formally, denote interim utility of buyer $i$ truthfully reporting $v_i$ as

$$u_i(v_i) := \mathop{\mathbb{E}}_{v_{-i}} [u_i(v_i, v_i, v_{-i})],$$

it is required that for all $v_i, \hat{v}_i$,

$$u_i(v_i) \geq \mathop{\mathbb{E}}_{v_{-i}} [u_i(v_i, \hat{v}_i, v_{-i})], \quad \text{and} \quad u_i(v_i) \geq 0.$$

For each buyer $i$, define the interim quantity

$$y_i(v_i) := \mathop{\mathbb{E}}_{v_{-i}} [x_i(v_i, v_{-i})^{\alpha}].$$

Bayesian incentive compatibility implies the envelope characterization

$$u_i'(v_i) = y_i(v_i), \qquad u_i(v_i) = u_i(0) + \int_0^{v_i} y_i(t)\, dt,$$

and, normalizing $u_i(0) = 0$, we obtain the following payment identity

$$\mathop{\mathbb{E}}_{v_{-i}} [p_i(v_i, v_{-i})] = v_i\, y_i(v_i) - \int_0^{v_i} y_i(t)\, dt. \tag{1}$$

The mechanism is BIC and IR if and only if $y_i(v_i)$ is non-decreasing in $v_i$, and the payment identity is satisfied. A complete derivation is provided in Appendix A.1.

*Remark* 2.1. As long as the ex-post allocation $x_i(v_i, v_{-i})$ is non-decreasing in $v_i$, one can strengthen the Bayesian incentive compatibility to dominant-strategy incentive compatibility (DSIC) using the payment rule determined by the ex-post payment identity:

$$p_i(v_i, v_{-i}) = v_i x_i(v_i, v_{-i})^{\alpha} - \int_0^{v_i} x_i(t, v_{-i})^{\alpha}\, dt.$$

**Seller's objective.** The seller designs the mechanism to maximize the expected profit

$$\mathrm{Prof} = \mathop{\mathbb{E}}_v \left[ \sum_{i=1}^n p_i(v) - c \max_{i \in [n]} x_i(v) \right].$$

Using (1) and integrating by parts yields the standard virtual welfare representation of expected profit:

$$\mathrm{Prof} = \mathop{\mathbb{E}}_v \left[ \sum_{i=1}^n \phi_i(v_i)\, x_i(v)^{\alpha} - c \max_{i \in [n]} x_i(v) \right], \tag{2}$$

where $\phi_i(v_i) = v_i - \frac{1 - F_i(v_i)}{f_i(v_i)}$ is Myerson's virtual value for buyer $i$. The derivation of (2) is provided in Appendix A.2.

Unlike classical multi-buyer auctions, allocations are not mutually exclusive in the data selling setting, and the production cost depends on the largest allocation regardless of the number of buyers served. This objective captures the central trade-off between expanding data production and the aggregate revenue generated across buyers.

**Optimal structure.** The form of the objective immediately implies a pooling structure. Intuitively, since the production cost depends only on the maximum allocation, to maximize the virtual welfare in (2), the maximal amount should be allocated to all buyers with positive virtual values, while buyers with negative virtual values receive no allocation. The common data amount depends on the sum of positive virtual values. When value distributions are regular, i.e., each virtual value function $\phi_i(v_i)$ is increasing, this yields a monotone allocation rule, which is indeed optimal. For irregular distributions, this approach holds by replacing the virtual values with the ironed virtual values $\phi_i^{\mathrm{ir}}(v_i)$ (Myerson, 1981). Concretely, let $R_i(q) := q \cdot F_i^{-1}(1-q)$, denote the revenue curve in quantile space. Let $\overline{R}_i$ be the smallest concave function that upper bounds $R_i$. Then the ironed virtual value is defined by

$$\phi_i^{\mathrm{ir}}(v) := \left. \frac{d}{dq}\overline{R}_i(q) \right|_{q=1-F_i(v)}$$

Before presenting the optimal mechanism, we introduce some notations used throughout the paper. Define notation $(z)_+ = \max\{z, 0\}$, and define $r_i$ as the positively-clipped ironed virtual value of each buyer $i$, that is,

$$r_i := \left(\phi_i^{\mathrm{ir}}(v_i)\right)_+.$$

**Theorem 2.2.** *In the optimal offline mechanism, all buyers with positive virtual values receive the same allocation. Specifically,*

$$D^* = \left(\frac{\alpha}{c}\sum_{i=1}^{n} r_i\right)^{\frac{1}{1-\alpha}} \quad and \quad x_i^* = \begin{cases} D^*, & r_i > 0, \\ 0, & otherwise. \end{cases}$$

Payments are determined by the allocation rule through the payment identity. The complete proof is provided in Appendix A.3. Notably, different buyers may be charged different payments for the same data level $D^*$ in the optimal mechanism, which depends on their value distribution.

We remark that the optimal mechanism can be extended to more general valuations in the form of $V_i(v_i, x_i) = v_i \cdot h_i(x_i)$, where $h_i(x_i)$ captures the exploitable return from data volume and is only assumed to be monotone. We provide an analysis of this generalization in Appendix A.4.

## 3. Online Data Production and Profit Maximization

We consider an online setting in which buyers arrive sequentially and the seller must make production and allocation decisions under demand uncertainty. This differs from the offline model studied in Section 2, where all buyer types are realized simultaneously and production can be matched exactly to demand.

### 3.1. Model and Preliminaries

There are $n$ buyers arriving one by one. Each buyer $i$ has a private type $v_i$, independently drawn from a known distribution $F_i$. The seller can produce data at constant marginal cost $c > 0$ and sell data to buyers. The seller commits ex ante to an online mechanism tailored to the distributions $\{F_i\}_{i=1}^n$. At each round $i$, after observing the reported type $\hat{v}_i$ and the current data level $D_{i-1}$, the mechanism determines production, allocation, and payment decisions.

**Online setting.** Initially, the available data amount is $D_0 = 0$. When buyer $i$ arrives, the seller decides: (i) how much additional data to produce, increasing the available amount from $D_{i-1}$ to $D_i \geq D_{i-1}$ at cost $c \cdot (D_i - D_{i-1})$, and (ii) how much data $x_i$ to allocate to buyer $i$, together with a payment $p_i$. Feasibility requires $x_i \leq D_i$.

At round $i$, all decisions depend only on the current data level $D_{i-1}$ and the reported type $\hat{v}_i$.

**Online mechanism.** An *online mechanism* is specified by functions $D_i(D_{i-1}, \hat{v}_i)$, $x_i(D_{i-1}, \hat{v}_i)$, and $p_i(D_{i-1}, \hat{v}_i)$ for $i = 1, \ldots, n$, which determine production, allocation, and payment at each round respectively.

Similar to offline model, if buyer $i$ has true type $v_i$ and reports $\hat{v}_i$ when previous data level is $D_{i-1}$, their utility is

$$u_i(v_i, \hat{v}_i; D_{i-1}) = v_i\, x_i(D_{i-1}, \hat{v}_i)^\alpha - p_i(D_{i-1}, \hat{v}_i).$$

The mechanism must satisfy incentive compatibility (IC) and individual rationality (IR), so that the seller may assume all reported values are truthful:

$$u_i(v_i, v_i; D_{i-1}) \geq u_i(v_i, \hat{v}_i; D_{i-1}),$$
$$u_i(v_i, v_i; D_{i-1}) \geq 0,$$

for all $v_i, \hat{v}_i$ and $D_{i-1} \in [0, +\infty)$.

**Seller's Objective.** The seller designs an online mechanism seeking to maximize expected profit,

$$\mathrm{Prof} := \mathbb{E}\left[\sum_{i=1}^{n} p_i - c\, D_n\right],$$

where the expectation is taken over buyer types $v_i \sim F_i$ and the induced production, allocation and payment decisions.

Similar to (2), we can rewrite the seller's expected profit with virtual value functions

$$\mathrm{Prof} = \mathbb{E}\left[\sum_{i=1}^{n} \phi_i(v_i)\, x_i^\alpha - c\, D_n\right].$$

## 3.2. Optimal Online Mechanism

We characterize the profit-optimal online mechanism through backward induction. The optimal policy takes the form of a dynamic program with a continuous state variable reflecting the current data level.

**Proposition 3.1** (Dynamic programming characterization)**.** *For $i \in [n]$ and $L \geq 0$, define $\Gamma_i(L)$ as the maximum expected profit obtainable from serving buyers $i, \cdots, n$, minus the expected total data production cost $cD_n$, under the constraint that $D_i \geq L$ (or equivalently, $D_{i-1} = L$). In particular, $\Gamma_1(0)$ is the expected profit of the optimal online mechanism, which we denote by*

$$\mathrm{ONL} := \Gamma_1(0).$$

*For any $i \in [n]$ and $L \geq 0$, $\Gamma_i(L)$ is recursively characterized by the following backward induction:*

$$\Gamma_i(L) = \mathop{\mathbb{E}}_{v_i \sim F_i}[\max_{D_i \geq L} r_i D_i^\alpha + \Gamma_{i+1}(D_i)],$$

*where we define $\Gamma_{n+1}(L) = -cL$ (i.e. the final total cost) for convenience. Recall that $r_i = (\phi_i^{\mathrm{ir}}(v_i))_+$.*

**Optimal production and allocation rule.** Given $r_i$ corresponding to reported value $v_i$, define function

$$\bar{D}_i(r_i) := \arg \max_{x \geq 0} r_i x^\alpha + \Gamma_{i+1}(x),$$

where the optimized objective is concave in $x$, and the maximum is always achieved at some finite $x$. $\bar{D}_i(r_i)$ denotes the optimal amount of data to produce at round $i$ if no data is carried over from the previous round, i.e., $D_{i-1} = 0$.

For $D_{i-1} \in [0, +\infty)$, the production and allocation in the optimal online mechanism for buyer $i$ are

$$D_i^*(D_{i-1}, v_i) = \max\{D_{i-1}, \bar{D}_i(r_i)\},$$

$$x_i^*(L, v_i) = \begin{cases} \max\{D_{i-1}, \bar{D}_i(r_i)\}, & r_i > 0, \\ 0, & r_i \leq 0. \end{cases}$$

That is, $\bar{D}_i(v_i)$ can be viewed as a target for data production at round $i$, such that the seller expands data production only when the virtual value is large enough so that $\bar{D}_i(r_i)$ exceeds the current data amount $L$.

The complete proof of Proposition 3.1 and the optimal policy is provided in Appendix B.1.

While the dynamic programming approach above provides a complete characterization of the profit-optimal online mechanism, the recursive definition of $\Gamma_i(L)$ involves nested optimizations over continuous state spaces, and no closed-form expression exists beyond restrictive settings. This not only limits its computational tractability, but also technically prevents us from establishing performance guarantees

through direct analysis of its expected profit. Moreover, the decision rules rely on full knowledge of value distributions and arrival order of buyers in the future, which further hinders its practical implementability.

Motivated by these considerations, we turn to study mechanisms with simple structure and analyze their profit performance, achieving a constant-factor approximation guarantee to the benchmark of optimal offline profit. This also implies a lower bound on the optimal online profit, enabling us to quantify the inherent loss due to online decision making.

## 3.3. Suboptimality of Simple Online Mechanisms

Before presenting a constant-factor approximation, we examine two natural classes of online mechanisms and show that neither achieves a constant approximation to the optimal online mechanism. These mechanisms serve as intuitive baselines that illustrate the limitations of online decision making in data markets.

**Myopic greedy mechanism.** The first baseline is the *myopic greedy* mechanism. Upon observing buyer $i$ with reported type $v_i$, the mechanism chooses the allocation that maximizes the instantaneous profit from this buyer alone, ignoring future buyers. Formally, the allocation is given by

$$x_i^{\mathrm{MG}}(v_i) := \arg \max_{x \geq 0}\{\phi_i^{\mathrm{ir}}(v_i)\, x^\alpha - cx\}$$

$$= \left(\tfrac{\alpha}{c} r_i\right)^{1/(1-\alpha)}.$$

The total data production after serving buyer $i$ is $D_i := \max\{D_{i-1}, x_i^{\mathrm{MG}}\}$.

The myopic greedy mechanism is optimal for a single buyer, but fails to account for the non-rivalry of data when multiple buyers are present.

**Theorem 3.2.** *There exists an instance with $n$ buyers such that the myopic greedy mechanism achieves at most an $O\left(n^{-\frac{\alpha}{1-\alpha}}\right)$ fraction of the optimal online profit.*

**Non-adaptive mechanism.** The second baseline is the *non-adaptive* mechanism. This mechanism commits ex ante to a fixed data production level and does not adjust production in response to realized buyer values. Formally, define

$$D_*^{\mathrm{NA}} := \arg \max_{x \geq 0} \mathbb{E}\left[\sum_{i=1}^n r_i x^\alpha - cx\right]$$

$$= \left(\tfrac{\alpha}{c} \mathbb{E}\left[\sum_{i=1}^n r_i\right]\right)^{\frac{1}{1-\alpha}}.$$

The data production level is fixed as $D_1 = D_2 = \cdots = D_n = D_*^{\mathrm{NA}}$, with allocation $x_i = \mathbb{I}[r_i > 0] \cdot D_*^{\mathrm{NA}}$.

The non-adaptive mechanism is optimal when buyer values are deterministic, but fails to respond to stochastic demand.

**Theorem 3.3.** *For any $\varepsilon > 0$, there exists an instance with $n \geq 2$ buyers such that any non-adaptive mechanism achieves at most an $\varepsilon$ fraction of the optimal online profit.*

### 3.4. Constant-Factor Approximation

Theorems 3.2 and 3.3 show that neither per-buyer optimization nor fixed production in advance suffices in online data markets with scaling-law valuations. These negative results motivate the need for online mechanisms that combine early data production with adaptive expansion. We design a simple two-stage mechanism that combines *initial expectation-based production* with *adaptive myopic greedy production*, which achieves a constant-factor approximation.

Our proposed mechanism adopts a two-stage structure, addressing *demand aggregation of buyers* and *adaptive response to stochastic values* respectively. In the first stage, the seller produces an initial amount of data, which only depends on the expected sum of positively-clipped virtual values, denoted

$$M := \sum_{i=1}^{n} \mathbb{E}[r_i].$$

In the second stage, the buyers arrive, and the data production is adaptively expanded. Similar to the myopic greedy mechanism, the adaptive data production in each round only optimizes for the current buyer. We will show that this simple structure is sufficient to achieve a constant-factor approximation to the offline optimum OFFL.

**Two-stage online mechanism.** The two-stage mechanism described in Algorithm 1 proceeds in two phases.

1. **Initial phase (before arrivals).** Compute

$$D_0 = \left( \frac{\alpha}{c} (1 - \delta) M \right)^{1/(1-\alpha)},$$

where $\delta \in (0, 1)$ is a constant parameter only depending on $\alpha$. A specific choice of $\delta$ is given in the proof.

2. **Online phase (upon arrivals).** When buyer $i$ reports $v_i$, compute $r_i = (\phi_i^{\text{ir}}(\hat{v}_i))_+$ and the myopic best-response

$$x_i^{\text{MG}} = \left( \frac{\alpha}{c} r_i \right)^{1/(1-\alpha)}.$$

If $x_i^{\text{MG}} > D_{i-1}$, produce additional data to raise the available amount to $D_i = x_i^{\text{MG}}$; otherwise keep $D_i = D_{i-1}$. Allocate $x_i = D_i$ to buyer $i$ if $\phi_i^{\text{ir}}(\hat{v}_i) \geq 0$, and the payment is computed according to Myerson's payment identity (1).

The pseudocode of the mechanism is presented in Algorithm 1 in Appendix B.4. Incentive compatibility follows by standard argument on allocation monotonicity.

**Implementability and approximation guarantee.** The two-stage online mechanism has a simpler structure than the optimal online mechanism, facilitating its practical implementability. The initial production only relies on a single aggregated expectation $M$, and the adaptive production in each round only depends on the current buyer's value and distribution. Nevertheless, it suffices to achieve a constant approximation to the optimal offline profit.

**Theorem 3.4.** *Given constant $\alpha \in (0, 1)$, taking $\delta = \frac{1-\alpha}{2-\alpha}$, for any instance, the two-stage online mechanism (Algorithm 1) achieves expected profit*

$$\text{ALG} \geq \frac{1}{(1 + \frac{1}{1-\alpha}) C_{\frac{1}{1-\alpha}}} \cdot \text{OFFL}$$

*where* ALG *denotes the expected profit of the two-stage online mechanism,* OFFL *denotes the expected profit of the optimal offline mechanism, and $C_{\frac{1}{1-\alpha}}$ denotes the best constant in Rosenthal's inequality for $L^{\frac{1}{1-\alpha}}$ space, which only depends on $\alpha$.*

The proof of this theorem utilizes Rosenthal's inequality, which is stated below. Applying this inequality with $X_i = r_i$, we express the optimal offline profit with the left-hand side of the inequality, while we show that the profit of Algorithm 1 approximates the right-hand side.

**Lemma 3.5** (Rosenthal (1970)). *Let $X_1, X_2, \ldots, X_n$ be independent non-negative random variables with $\mathbb{E}[X_i^p] < \infty$ for some $p \geq 1$. Then there exists a constant $C_p > 0$, depending only on $p$, such that*

$$\mathbb{E}\left[ \left( \sum_{i=1}^{n} X_i \right)^p \right] \leq C_p \max \left\{ \sum_{i=1}^{n} \mathbb{E}[X_i^p], \left( \sum_{i=1}^{n} \mathbb{E}X_i \right)^p \right\}.$$

*Specifically, the best constant $C_p$ is given in (Ibragimov & Sharakhmetov, 2001) as follows: $C_p = 2$, for $1 < p < 2$; $C_p = \mathbb{E}_{Z \sim \text{Poisson}(1)}[Z^p]$, for $p \geq 2$. Asymptotically, $C_p = \Theta((\frac{p}{\log p})^p)$.*

**Online-offline profit gap.** Since the profit of the two-stage mechanism is a lower bound on the optimal online profit, as an immediate corollary of Theorem 3.4, we have that $\frac{\text{ONL}}{\text{OFFL}} \geq \frac{1}{(1+\frac{1}{1-\alpha}) C_{\frac{1}{1-\alpha}}}$ holds for all instances. That is, the optimal online profit is a constant approximation to the optimal offline profit given constant $\alpha \in (0, 1)$. In complement, we show an upper bound on the worst-case ratio between the optimal online and offline profits. Tecnically, to prove this upper bound, we analyze hard instances where each buyer's virtual value follows a $\text{Bernoulli}(\frac{1}{n})$ distribution. The binary nature of the virtual values will enable a

closed-form analysis on the optimal online algorithm following Proposition 3.1. Moreover, in the limit case that $n \to +\infty$, the number of buyers with positive virtual values will become a $\mathrm{Poisson}(1)$ variable, closely related to the best constant in Rosenthal's inequality.

**Theorem 3.6.** *Given $\alpha \in (0,1)$, for sufficiently large $n$, there exists an instance with $n$ buyers such that*

$$\frac{\mathrm{ONL}}{\mathrm{OFFL}} \leq \frac{2^{\frac{1}{1-\alpha}}}{C_{\frac{1}{1-\alpha}}} = e^{-\Theta(\frac{1}{1-\alpha} \log \frac{1}{1-\alpha})}.$$

*where $C_{\frac{1}{1-\alpha}}$ is the best constant in Rosenthal's inequality.*

Note that when the elasticity parameter $\alpha$ is close to 1, the constant $C_{\frac{1}{1-\alpha}}$ from Rosenthal's inequality is asymptotically $C_{\frac{1}{1-\alpha}} = \exp(\Theta(\frac{1}{1-\alpha} \log \frac{1}{1-\alpha}))$, which is super-exponential in $\frac{1}{1-\alpha}$. As a result, $C_{\frac{1}{1-\alpha}}$ is the dominant factor in the approximation ratios in both Theorem 3.4 and Theorem 3.6, which demonstrates that the requirement of online decision-making inherently incurs a worst-case profit ratio between the optimal online and offline profits that is super-exponential in $\frac{1}{1-\alpha}$. This also suggests that the simple two-stage online mechanism is nearly optimal within the class of an online mechanism.

# 4. Gain-from-Trade Maximization in Bilateral Data Trading

In this section, we study a bilateral data trading scenario between a seller and a buyer. Unlike the previous sections, the seller is now a strategic agent with private information, and the mechanism is designed by a third-party mediator to maximize social welfare. Although buyer valuations follow the same scaling-law structure, this shift leads to a fundamentally different mechanism design problem.

## 4.1. Model

There is a single buyer and a single seller. The buyer has a private value $v > 0$, drawn from a known distribution $F^B$ with density $f^B$. The seller has a private unit cost $c > 0$, drawn independently from a known distribution $F^S$ with density $f^S$. Both first report their types to a mediator, who then determines the allocation and payments. If the mechanism allocates $x \geq 0$ units of data, the buyer's valuation is $vx^\alpha$, where $\alpha \in (0,1)$ is the elasticity parameter, and the seller's cost is $cx$.

A bilateral-trading mechanism specifies an allocation rule $x(v,c) \geq 0$, a payment $p^B(v,c) \geq 0$ charged to the buyer, and a payment $p^S(v,c) \geq 0$ paid to the seller. The buyer's utility when reporting $\hat{v}$ is

$$u^B(v,\hat{v};c) = v\,x(\hat{v},c)^\alpha - p^B(\hat{v},c),$$

and the seller's utility reporting $\hat{c}$ is

$$u^S(c,\hat{c};v) = p^S(v,\hat{c}) - c\,x(v,\hat{c}).$$

The mechanism is required to be Bayesian incentive compatible and individually rational for both agents. Additionally, it is constrained by weak budget balance in expectation, i.e.

$$\mathbb{E}[p^B(v,c) - p^S(v,c)] \geq 0.$$

The mediator's objective is to maximize the expected gain-from-trade (GFT),

$$\mathrm{GFT} = \mathbb{E}_{v \sim F^B, c \sim F^S}[v\,x(v,c)^\alpha - c\,x(v,c)].$$

**Virtual-value representation.** Under truthfulness, expected payments admit a virtual-value representation. Define the buyer's virtual value $\phi^B(v) = v - \frac{1-F^B(v)}{f^B(v)}$, and the seller's virtual cost $\phi^S(c) = c + \frac{F^S(c)}{f^S(c)}$. Then expected payments satisfy

$$\mathbb{E}[p^B(v,c)] = \mathbb{E}\left[\phi^B(v)\,x(v,c)^\alpha\right], \qquad (3)$$

$$\mathbb{E}[p^S(v,c)] = \mathbb{E}\left[\phi^S(c)\,x(v,c)\right]. \qquad (4)$$

A complete derivation of the payment identities on both the buyer and seller sides is provided in Appendix C.1.

## 4.2. First-Best and Second-Best Benchmarks

**First-best.** If $v$ and $c$ are publicly known, the welfare-maximizing allocation solves

$$\max_{x \geq 0}\{vx^\alpha - cx\},$$

with solution $x^*(v,c) = (\alpha v/c)^{1/(1-\alpha)}$. This benchmark ignores incentive and budget-balance constraints and serves as an upper bound on the achievable gain-from-trade. The resulting first-best GFT is

$$\mathrm{FB} = \mathbb{E}\left[(1-\alpha)\alpha^{\frac{\alpha}{1-\alpha}} \cdot v^{\frac{1}{1-\alpha}} \cdot c^{-\frac{\alpha}{1-\alpha}}\right].$$

**Second-best.** Among mechanisms that are truthful, individually rational, and weakly budget balanced, the optimal (second-best) mechanism can be characterized via a Lagrangian relaxation. There exists a multiplier $\lambda^* \geq 0$ such that the second-best allocation is

$$x^{\mathrm{SB}}(v,c) = \left(\frac{\alpha\left(v + \lambda^*\phi^B(v)\right)}{c + \lambda^*\phi^S(c)}\right)^{\frac{1}{1-\alpha}},$$

whenever the numerator and denominator are positive, and 0 otherwise, interpreted using ironed virtual values when necessary. A formal derivation of the second-best mechanism is given in Appendix C.2.

While the second-best mechanism is optimal among all truthful and budget-balanced mechanisms, its dependence on the endogenous multiplier $\lambda^*$ obscures its structure and complicates comparison with the first-best benchmark or with the outcome of other mechanisms. Motivated by this, we next study simpler mechanisms that admit direct comparison with the first-best gain from trade and achieve explicit constant-factor approximation guarantees.

### 4.3. Constant-Factor Approximation to First-Best GFT

We now turn to two simple, structurally transparent and natural mechanisms: *seller-proposing* and *buyer-proposing* mechanisms, each of which grants proposal power to one side of the market. Both mechanisms are truthful and individually rational by construction. In the classical single-item bilateral trading setting, such as (Deng et al., 2022), the two mechanisms must be carefully combined to achieve a constant fraction of first-best GFT. As a critical distinction, in our setting governed by scaling law, either of these two mechanisms alone suffices to guarantee constant approximation to first-best GFT, given constant $\alpha \in (0, 1)$. In particular, the buyer-proposing mechanism achieves a $1/e$ approximation ratio for any $\alpha \in (0, 1)$.

**Seller-proposing mechanism (SellerP).** In the seller-proposing mechanism, the seller receives payment from the buyer directly, and the seller proposes an allocation that maximizes her obtained virtual surplus based on buyer's virtual value. Formally, the allocation rule is

$$x^{\mathrm{SP}}(v, c) = \left( \frac{\alpha\, \phi^B(v)}{c} \right)_+^{\frac{1}{1-\alpha}}.$$

Intuitively, the seller internalizes the buyer's information through the virtual value transformation, while treating her own production cost directly. We assume $F^B$ is regular for simplicity. For irregular distributions, $\phi^B(v)$ is replaced by its ironed version, and the same applies to the buyer-proposing mechanism when $F^S$ is irregular. All results continue to hold.

**Buyer-proposing mechanism (BuyerP).** Symmetrically, in the buyer-proposing mechanism, the buyer pays to the seller directly, and the buyer proposes an allocation that maximizes her obtained virtual surplus based on the seller's virtual cost. The resulting allocation rule is

$$x^{\mathrm{BP}}(v, c) = \left( \frac{\alpha\, v}{(\phi^S(c))_+} \right)^{\frac{1}{1-\alpha}},$$

where $\phi^S(c)$ is the seller's virtual cost. Here the buyer responds directly to her own valuation, while incorporating the seller's information through the virtual cost transformation.

**Decomposition of welfare.** A key insight underlying our analysis is that scaling-law valuations induce a multiplicative decomposition of first-best welfare:

$$\mathrm{FB} = (1-\alpha)\alpha^{\frac{\alpha}{1-\alpha}} \cdot \underset{v\sim F^B}{\mathbb{E}}\left[v^{\frac{1}{1-\alpha}}\right] \cdot \underset{c\sim F^S}{\mathbb{E}}\left[c^{-\frac{\alpha}{1-\alpha}}\right].$$

This decomposition isolates buyer-side and seller-side contributions to welfare. Moreover, both the seller's utility in seller-proposing mechanisms and the buyer's utility in buyer-proposing mechanism can be similarly decomposed, each replacing the term of buyer's value or seller's cost with the virtual value or virtual cost, respectively. This enables a clear analysis on their approximation ratio. A formal statement of this property is presented in Lemma C.1 in Appendix C.3.

**Approximation guarantees.** Utilizing this decomposition property, we obtain constant-factor approximation guarantees for both mechanisms.

**Theorem 4.1.** *The Seller-proposing mechanism achieves an $\alpha^{\frac{1}{1-\alpha}}$-approximation to the first-best gain-from-trade.*

**Theorem 4.2.** *The Buyer-proposing mechanism achieves an $\alpha^{\frac{\alpha}{1-\alpha}}$-approximation to the first-best gain-from-trade, and in particular at least an $1/e$-approximation.*

As an immediate corollary, the second-best GFT is also at least $1/e$ of the first-best GFT. Our approach differs from that used in existing bilateral trade approximation results. The proofs exploit the above welfare decomposition technique, and reduce the approximation analysis to inequalities for one-dimensional integrals on the buyer or seller side. Complete proofs are provided in Appendix C.4 and C.5.

These results show that even under bilateral private information and truthfulness constraints, a substantial fraction of the first-best welfare can be recovered by mechanisms with very simple and transparent structure. Moreover, the approximation factors depend only on the scaling parameter $\alpha$ and not on the underlying distributions. This highlights how the scaling-law property of data valuations enables robust, interpretable welfare guarantees and provides a theoretical foundation for using simple market rules in data trading environments.

## 5. Discussion and Future Work

We studied market design for data trading under scaling-law valuations, focusing on how the non-rivalrous nature of data and diminishing returns to scale fundamentally alter classical mechanism design insights. Across offline, online, and bilateral trading settings, we showed that simple mechanisms with transparent structure can achieve strong performance guarantees, including constant-factor approximations to natural welfare benchmarks. Our results in the offline setting also extend to more general valuation models.

## 5.1. Modeling Choices and Limitations

Our analysis operates under several modeling assumptions, which define the scope of the current model and highlight directions for future generalization.

- We adopt a power-law functional form for buyer valuations, motivated by scaling laws for large model training (Kaplan et al., 2020; Hoffmann et al., 2022), which indicate a power-law relationship between dataset size and model performance.

- We assume the scaling exponent $\alpha$ is identical across buyers and publicly known, which enables us to isolate the role of private value $v_i$.

- We treat data as a homogeneous good, so a buyer's utility depends only on the quantity received. Consequently, the seller's cost depends only on the maximum allocated dataset size.

These assumptions align with important practical scenarios, particularly in the training of foundation models where distinct architectures frequently follow analogous scaling trajectories. Moreover, they are standard assumptions in economic and game-theoretic literature, facilitating rigorous analytical tractability.

At the same time, these assumptions may not capture complex scenarios with stronger heterogeneity, and consequently, the approximation guarantees derived here may not carry over to those richer environments without new analytical tools. Recognizing these boundaries helps situate our contributions and motivates the open questions discussed below.

## 5.2. Future Directions

Several directions for future work remain open.

**Relaxing modeling assumptions.** Extending the theoretical framework to accommodate general valuation or cost functions, heterogeneous or private scaling exponents, heterogeneous data attributes, and more complex market structures would substantially broaden the applicability of our results. Analytical challenges may arise due to more complicated decision structures and information structures, necessitating different technical approaches.

**Empirical and computational evaluation.** While this paper primarily focuses on theoretical analysis providing worst-case guarantees and structural characterizations, empirical validation on synthetic or real-world data distributions remains a valuable future direction. Such investigations would illuminate the average-case performance and practical efficacy of the proposed mechanisms.

**Unknown priors and dynamic learning.** Our mechanism design framework assumes known prior distributions over valuations. A further direction is to incorporate online learning of unknown underlying distributions of demands and valuations.

## Acknowledgements

This research was supported by the National Natural Science Foundation of China (Grant No. 62572010), China Unicom Software Research Institute, and Beijing National Science Foundation (Grant No. QY25120). Zihe Wang was supported by the National Natural Science Foundation of China (Grant No. 62172422). We thank the anonymous reviewers of ICML 2026, and the anonymous reviewers of ICLR 2026 AIMS Workshop, for their thoughtful feedback and helpful suggestions.

## Impact Statement

This paper is primarily theoretical and contributes to the theory of mechanism design. While the results may have potential societal implications, the paper itself focuses on foundational questions and does not introduce immediate real-world deployments that require separate discussion here.

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

# Appendix

## A. Missing Proofs in Section 2

### A.1. Envelope Characterization for Scaling-Law Utilities

In this section, we derive the envelope and payment identities used in Section 2. Consider a direct-revelation mechanism $(x(\cdot), p(\cdot))$ with type profile $v = (v_1, \ldots, v_n)$, where buyer $i$ receives allocation $x_i(v) \geq 0$ and pays $p_i(v)$. Fix a buyer $i$ and fix $v_{-i}$. If buyer $i$'s true type is $v_i$ and she reports $\hat{v}_i$, her (ex-post) utility is

$$u_i(v_i, \hat{v}_i; v_{-i}) = v_i \, x_i(\hat{v}_i, v_{-i})^\alpha - p_i(\hat{v}_i, v_{-i}).$$

Define the interim allocation and interim payment (as functions of buyer $i$'s report)

$$y_i(v_i) := \mathop{\mathbb{E}}_{v_{-i}} \left[ x_i(v_i, v_{-i})^\alpha \right], \qquad \tilde{p}_i(v_i) := \mathop{\mathbb{E}}_{v_{-i}} \left[ p_i(v_i, v_{-i}) \right],$$

and the interim truthful utility

$$U_i(v_i) := \mathop{\mathbb{E}}_{v_{-i}} \left[ v_i \, x_i(v_i, v_{-i})^\alpha - p_i(v_i, v_{-i}) \right] = v_i y_i(v_i) - \tilde{p}_i(v_i).$$

Bayesian incentive compatibility (BIC) implies that for all $v_i, \hat{v}_i \geq 0$,

$$U_i(v_i) \geq v_i y_i(\hat{v}_i) - \tilde{p}_i(\hat{v}_i).$$

Applying this inequality twice (once with $(v_i, \hat{v}_i)$ and once with $(\hat{v}_i, v_i)$), for any $v_i > v'_i$ we obtain

$$(v_i - v'_i) \, y_i(v'_i) \leq U_i(v_i) - U_i(v'_i) \leq (v_i - v'_i) \, y_i(v_i).$$

In particular, $y_i(\cdot)$ is nondecreasing, and $U_i$ is absolutely continuous with $U'_i(v_i) = y_i(v_i)$ for almost every $v_i$. Normalizing $U_i(0) = 0$ (which is without loss under interim IR), we get the envelope formula

$$U_i(v_i) = \int_0^{v_i} y_i(t) \, dt.$$

Substituting $U_i(v_i) = v_i y_i(v_i) - \tilde{p}_i(v_i)$ yields the interim payment identity

$$\tilde{p}_i(v_i) = v_i \, y_i(v_i) - \int_0^{v_i} y_i(t) \, dt,$$

which is (1).

### A.2. Derivation of Expected Revenue and Profit

In this section, we derive the virtual-value representation (2). From (1), for each buyer $i$ we have

$$\mathbb{E}[p_i(v)] = \mathop{\mathbb{E}}_{v_i}[v_i y_i(v_i)] - \mathop{\mathbb{E}}_{v_i}\left[ \int_0^{v_i} y_i(t) \, dt \right].$$

Writing the expectations as integrals (with $v_i \sim F_i$ and density $f_i$),

$$\mathop{\mathbb{E}}_{v_i}[v_i y_i(v_i)] = \int_0^\infty v_i y_i(v_i) f_i(v_i) \, dv_i,$$

and by Fubini's theorem,

$$\mathop{\mathbb{E}}_{v_i}\left[ \int_0^{v_i} y_i(t) \, dt \right] = \int_0^\infty y_i(t) \left( 1 - F_i(t) \right) dt = \int_0^\infty y_i(v_i) \frac{1 - F_i(v_i)}{f_i(v_i)} f_i(v_i) \, dv_i.$$

Therefore,

$$\mathbb{E}[p_i(v)] = \int_0^\infty \left( v_i - \frac{1 - F_i(v_i)}{f_i(v_i)} \right) y_i(v_i) \, f_i(v_i) \, dv_i.$$

Define buyer $i$'s (Myerson) virtual value

$$\phi_i(v_i) \; := \; v_i - \frac{1 - F_i(v_i)}{f_i(v_i)}.$$

Using $y_i(v_i) = \mathbb{E}_{v_{-i}}[x_i(v_i, v_{-i})^\alpha]$ and the law of iterated expectation, we obtain the standard virtual-surplus form:

$$\mathbb{E}[p_i(v)] \; = \; \mathbb{E}_v[\phi_i(v_i)\, x_i(v)^\alpha].$$

Summing over buyers and subtracting the production cost (in the offline model, the seller produces $D = \max_i x_i(v)$ and pays cost $cD$) gives

$$\mathbb{E}[\mathrm{Prof}] = \mathbb{E}_v\left[\sum_{i=1}^n \phi_i(v_i)\, x_i(v)^\alpha - c \max_{i \in [n]} x_i(v)\right],$$

which is (2).

### A.3. Proof of Theorem 2.2

*Proof.* By (2), the seller's objective is to maximize

$$\mathrm{Prof} = \mathbb{E}_v\left[\sum_{i=1}^n \phi_i(v_i)\, x_i(v)^\alpha \; - \; c \max_{i \in [n]} x_i(v)\right].$$

For possibly irregular distributions, by (Myerson, 1981), the virtual values can be replaced by ironed virtual values under allocation monotonicity, that is,

$$\mathrm{Prof} = \mathbb{E}_v\left[\sum_{i=1}^n \phi_i^{\mathrm{ir}}(v_i)\, x_i(v)^\alpha \; - \; c \max_{i \in [n]} x_i(v)\right].$$

**Reduction to a single decision variable.** Fix a realized type profile $(v_1, \ldots, v_n)$. Let $S = \{i : \phi_i^{\mathrm{ir}}(v_i) > 0\}$ denote the set of buyers with positive virtual values. For any feasible allocation, decreasing $x_i$ for buyers with $\phi_i^{\mathrm{ir}}(v_i) \le 0$ weakly increases profit, so in any optimal solution $x_i = 0$ for all $i \notin S$.

For buyers in $S$, suppose the seller produces a dataset of size $D$ and allocates $x_i \le D$ to each $i \in S$. Since the virtual surplus term is increasing in $x_i$ for $\phi_i^{\mathrm{ir}}(v_i) > 0$, optimality requires $x_i = D$ for all $i \in S$. Hence the problem reduces to choosing $D \ge 0$ to maximize

$$\sum_{i \in S} \phi_i^{\mathrm{ir}}(v_i)\, D^\alpha - cD,$$

which equals

$$\sum_{i \in [n]} r_i\, D^\alpha - cD,$$

where $r_i = \max\{\phi_i^{\mathrm{ir}}(v_i), 0\}$.

**Optimal production level.** The objective is concave in $D$. The first-order condition is

$$\alpha D^{\alpha-1} \sum_{i \in S} r_i = c,$$

which yields the unique maximizer

$$D^* = \left(\frac{\alpha}{c} \sum_{i \in S} r_i\right)^{\frac{1}{1-\alpha}}.$$

Substituting back gives the allocation stated in Theorem 2.2.

**Payments.** Given the allocation rule, payments are determined by the Bayesian envelope formula. For each buyer $i$,

$$p_i(v_i, v_{-i}) = v_i x_i(v_i, v_{-i})^\alpha - \int_0^{v_i} x_i(t, v_{-i})^\alpha \, dt,$$

This completes the proof. □

### A.4. Extension to Generalized Valuation

In this section, we consider a generalized form of buyers' valuation for allocated data. For each buyer $i \in [n]$, suppose her valuation given allocation $x_i \geq 0$ is

$$V_i(v_i, x_i) = v_i \cdot h_i(x_i),$$

where $h_i(x_i)$ represents the exploitable return of data volume. If $h_i(x_i) = x_i^\alpha$, it recovers our main setting. We assume that $h_i(x_i)$ is a publicly known function which is continuous and non-decreasing in $x_i$, and assume an eventually diminishing return, that is, $\lim_{x_i \to \infty} \frac{h_i(x_i)}{x_i} = 0$.

We generalize the optimal mechanism in Theorem 2.2 to this setting.

**Theorem A.1.** *In the optimal offline mechanism, all buyers with positive ironed virtual values (i.e. $r_i \phi_i^{\mathrm{ir}}(v_i) > 0$) receive the same allocation $D^*$. Specifically,*

$$D^*(v) = \arg\max_{D \geq 0} \sum_{i=1}^n r_i \cdot h_i(D) - c\,D,$$

*and*

$$x_i(v) = \begin{cases} D^*(v), & r_i > 0, \\ 0, & \text{otherwise.} \end{cases}$$

*Proof.* Similar to the proof of Theorem 2.2, the optimal mechanism maximizes

$$\mathrm{Prof} = \mathbb{E}_v \left[ \sum_{i=1}^n \phi_i^{\mathrm{ir}}(v_i)\, h_i(x_i(v)) - c \max_{i \in [n]} x_i(v) \right].$$

Fix a realized type profile $(v_1, \ldots, v_n)$. Let $S = \{i : \phi_i^{\mathrm{ir}}(v_i) > 0\}$ denote the set of buyers with positive virtual values. For any feasible allocation, decreasing $x_i$ for buyers with $\phi_i^{\mathrm{ir}}(v_i) \leq 0$ weakly increases profit, so in any optimal solution $x_i = 0$ for all $i \notin S$.

For buyers in $S$, suppose the seller produces a dataset of size $D$ and allocates $x_i \leq D$ to each $i \in S$. Since the virtual surplus term is increasing in $x_i$ for $\phi_i^{\mathrm{ir}}(v_i) > 0$, optimality requires $x_i = D$ for all $i \in S$. Hence the problem reduces to choosing $D \geq 0$ to maximize

$$\sum_{i \in S} \phi_i^{\mathrm{ir}}(v_i)\, h_i(D) - cD,$$

which equals

$$\sum_{i \in [n]} r_i\, h_i(D) - cD,$$

where $r_i = \max\{\phi_i^{\mathrm{ir}}(v_i), 0\}$.

**Optimal production level.** Although the objective is not necessarily concave in $D$, by the assumption of eventually diminishing return and finiteness of virtual values, we have

$$\lim_{D \to \infty} \sum_{i \in [n]} r_i\, h_i(D) - cD = -\infty.$$

Therefore, there exists a finite optimal solution $D^* = \arg\max_{D \geq 0} \sum_{i \in [n]} r_i\, h_i(D) - cD$. If there are multiple optimal solutions, take the smallest one, which exists by continuity.

**Allocation monotonicity.** Now we prove that this yields a monotone allocation rule. It suffices to show that $D^*(v)$ is weakly increasing in each $v_i$. Suppose for contradiction that $D^*(v_i, v_{-i}) > D^*(v'_i, v_{-i})$ for some $v_i < v'_i$. Define $D_1 = D^*(v_i, v_{-i})$, $D_2 = D^*(v'_i, v_{-i})$, and

$$H_1(D) = \sum_{j \in [n]} (\phi_j^{\text{ir}}(v_j))_+ h_j(D),$$

$$H_2(D) = (\phi_i^{\text{ir}}(v'_i))_+ h_i(D) + \sum_{j \in [n] \setminus \{i\}} (\phi_j^{\text{ir}}(v_j))_+ h_j(D).$$

Then we have

$$H_1(D_1) - cD_1 > H_1(D_2) - cD_2, \quad H_2(D_1) - cD_1 \leq H_2(D_2) - cD_2,$$

where the first inequality is strict because the tie-breaking would be in favor of $D_2$.

By subtraction, we have

$$H_1(D_1) - H_2(D_1) > H_1(D_2) - H_2(D_2).$$

That is,

$$((\phi_i^{\text{ir}}(v_i))_+ - (\phi_i^{\text{ir}}(v'_i))_+)(h_i(D_1) - h_i(D_2)) > 0$$

We have $h_i(D_1) \geq h_i(D_2)$ by the monotonicity of $h_i$. Also, since $v_i < v'_i$, we have $(\phi_i^{\text{ir}}(v_i))_+ - (\phi_i^{\text{ir}}(v'_i))_+ \leq 0$. It follows that $((\phi_i^{\text{ir}}(v_i))_+ - (\phi_i^{\text{ir}}(v'_i))_+)(h_i(D_1) - h_i(D_2)) \leq 0$, which contradicts.

Therefore, the resulting allocation rule is monotone, and the mechanism satisfies incentive compatibility and individual rationality with payment rule determined by the payment identity.

$\square$

# B. Missing Proofs in Section 3

## B.1. Proof of Proposition 3.1

*Proof.* We prove the dynamic programming characterization by backward induction. Recall that for each round $i \in \{1, \ldots, n+1\}$ and current data level $L \geq 0$ (i.e. $D_{i-1} = L$), define $\Gamma_i(L)$ as the maximum expected profit from buyers $i, \ldots, n$ minus the total production cost $cD_n$, given that the data level at the start of round $i$ is $L$. The boundary condition and recursion are derived as follows.

**Base case ($i = n + 1$):** After all $n$ buyers have been served, no further payments are collected. The total production cost is $c$ times the final data level $D_n$. Given that the data level at the start of round $n+1$ is $L$ (i.e., $D_n = L$), the profit from round $n+1$ onward is $-cL$. Thus,

$$\Gamma_{n+1}(L) = -cL.$$

**Inductive step ($i \leq n$):** Assume $\Gamma_{i+1}(L)$ correctly represents the maximum expected future profit from rounds $i+1$ to $n$ minus $cD_n$, given initial data level $L$ at round $i+1$. At round $i$, the initial data level is $L = D_{i-1}$. The seller observes buyer $i$'s reported type $\hat{v}_i$. By incentive compatibility (IC), we may assume truthful reporting ($\hat{v}_i = v_i \sim F_i$). The seller chooses a production level $D_i \geq L$ and determines allocation $x_i$ and payment $p_i$.

By the virtual welfare representation (analogous to Section 2), under IC and individual rationality (IR), the expected payment from buyer $i$ satisfies

$$\mathbb{E}[p_i \mid v_i, D_i] = \mathbb{E}\left[r_i x_i^{\alpha} \mid v_i, D_i\right],$$

where $r_i = (\phi_i^{\text{ir}}(v_i))_+$ is the non-negative ironed virtual value, and $\phi_i^{\text{ir}}$ is the ironed virtual value function ensuring monotonicity of the allocation rule. For fixed $D_i$, the optimal allocation is $x_i = D_i$ if $r_i > 0$ and $x_i = 0$ otherwise, which is monotone in $v_i$ due to the ironing procedure. This yields an immediate expected virtual surplus of $r_i D_i^{\alpha}$.

The data is non-rivalrous, so the post-production data level $D_i$ carries forward to round $i+1$ unchanged by allocation. The expected future profit from round $i+1$ to $n$ is $\Gamma_{i+1}(D_i)$. The seller maximizes the sum of immediate virtual surplus and future profit over $D_i \geq L$:

$$\max_{D_i \geq L} \left\{ r_i D_i^{\alpha} + \Gamma_{i+1}(D_i) \right\}.$$

Taking expectation over $v_i \sim F_i$,

$$\Gamma_i(L) = \mathbb{E}_{v_i \sim F_i} \left[ \max_{D_i \geq L} \{r_i D_i^\alpha + \Gamma_{i+1}(D_i)\} \right].$$

**Finiteness and concavity:** To show that $\Gamma_i(L)$ is well-defined, we prove that the maximized objective $H_i(x) := r_i x^\alpha + \Gamma_{i+1}(x)$ is concave in $x$ for each realization of $r_i$, and is maximized at finite $x$. Concavity follows by induction: $\Gamma_{n+1}(x) = -cx$ is concave, and if $\Gamma_{i+1}$ is concave, then $H_i(x)$ is concave as a non-negative weighted sum of concave functions ($x^\alpha$ is concave for $\alpha \in (0, 1]$ and $\Gamma_{i+1}$ concave by hypothesis). Since $c > 0$ and virtual values are bounded, $H_i(x) \to -\infty$ as $x \to \infty$, ensuring a finite maximizer. Thus, $\bar{D}_i(r_i) = \arg\max_{x \geq 0} H_i(x)$ exists and is finite.

**Optimal policy:** Given $L$ and $v_i$, the optimal production level maximizes $H_i(x_i)$ subject to $x_i \geq L$. By concavity,

$$D_i^*(L, v_i) = \max\left\{ L, \bar{D}_i(r_i) \right\}.$$

The allocation rule is

$$x_i^*(L, v_i) = \begin{cases} D_i^*(L, v_i) & \text{if } r_i > 0, \\ 0 & \text{if } r_i = 0, \end{cases}$$

which ensures IC and IR. The payment $p_i$ is set via the envelope theorem to satisfy IC and extract surplus. This policy is optimal by construction of $\Gamma_i(L)$.

Finally, $\Gamma_1(0)$ is the expected profit of the optimal online mechanism, denoted ONL. $\qquad\square$

### B.2. Proof of Theorem 3.2

*Proof.* Consider an instance with $n$ buyers. For each $i$, let $F_i$ be the degenerate distribution at $v_i = 1$. Thus, $r_i = (\phi_i(v_i))_+ = 1$ for all $i$.

**Profit of the myopic greedy mechanism.** The myopic greedy mechanism solves, in each round $i$,

$$x_i^{\text{MG}} = \arg\max_{x \geq 0}\{x^\alpha - cx\} = \left(\tfrac{\alpha}{c}\right)^{\frac{1}{1-\alpha}}.$$

Each buyer receives this amount of data, so the total virtual surplus equals

$$\sum_{i=1}^{n} x^\alpha = n\left(\tfrac{\alpha}{c}\right)^{\frac{\alpha}{1-\alpha}}.$$

Since production is shared, the total cost equals

$$c\max_i x_i = c\left(\tfrac{\alpha}{c}\right)^{\frac{1}{1-\alpha}} = \alpha\left(\tfrac{\alpha}{c}\right)^{\frac{\alpha}{1-\alpha}}.$$

Hence the greedy profit is

$$\text{ALG} = (n - \alpha)\left(\tfrac{\alpha}{c}\right)^{\frac{\alpha}{1-\alpha}}.$$

**Optimal online profit.** The optimal online mechanism aggregates demand across all buyers and the optimal production level is

$$D^* = \arg\max_{D \geq 0}\{nD^\alpha - cD\} = \left(\tfrac{n\alpha}{c}\right)^{\frac{1}{1-\alpha}},$$

yielding profit

$$\text{ONL} = (1 - \alpha)n^{\frac{1}{1-\alpha}}\left(\tfrac{\alpha}{c}\right)^{\frac{\alpha}{1-\alpha}}.$$

**Approximation ratio.** The ratio of profit between myopic greedy mechanism to optimal online profit is therefore

$$\frac{\text{ALG}}{\text{ONL}} = \frac{n - \alpha}{(1 - \alpha)n^{\frac{1}{1-\alpha}}} = O\left(n^{-\frac{\alpha}{1-\alpha}}\right),$$

which proves the theorem. $\qquad\square$

### B.3. Proof of Theorem 3.3

*Proof.* Let $H > 1$ be a sufficiently large constant to be determined later. Construct the instance such that each buyer has virtual value $\phi_i(v_i) = H$ with probability $\frac{1}{nH}$ and $\phi_i(v_i) = 0$ otherwise. Specifically, each buyer's value $v_i$ follows the truncated equal-revenue distribution with CDF $F_i(v_i) = \begin{cases} 1, & v_i \geq H \\ \max\{0, 1 - \frac{1}{nv_i}\}, & v_i \in [0, H) \end{cases}$.

**Expected profit of a non-adaptive mechanism.** Consider any non-adaptive mechanism that commits to a production level $D$ before observing buyer values. The expectation of each buyer's virtual value is $\frac{1}{n}$. Thus, the expected virtual surplus given production level $D$ is

$$\mathbb{E}\left[\sum_i \phi_i(v_i)\right] D^\alpha = D^\alpha.$$

And the expected profit is

$$D^\alpha - cD.$$

This is maximized when taking $D = D_*^{NA} = \left(\frac{\alpha}{c}\right)^{\frac{1}{1-\alpha}}$, which yields the optimal profit of any non-adaptive mechanism:

$$\mathrm{ALG} = (1 - \alpha)\left(\frac{\alpha}{c}\right)^{\frac{\alpha}{1-\alpha}}.$$

**Optimal online benchmark.** Consider an adaptive online mechanism that produces data with a fixed amount of

$$D^* = \left(\frac{H\alpha}{c}\right)^{\frac{1}{1-\alpha}}$$

when it sees the first buyer with $\phi_i(v_i) = H$, and sells the data to all buyers with $\phi_i(v_i) = H$. Otherwise, it produces $0$ amount of data when all buyers' virtual values are zero. Here $D^*$ is selected to maximize the expected profit based on the value distributions.

With probability $1 - (1 - \frac{1}{nH})^n \geq \frac{1}{H}$, at least one buyer has virtual value $H$. Conditioning on this event, the adaptive online mechanism's expected profit is at least

$$HD^{*\alpha} - cD^* = (1 - \alpha)\left(\frac{\alpha}{c}\right)^{\frac{\alpha}{1-\alpha}} H^{\frac{1}{1-\alpha}}.$$

And when this event does not happen, the online mechanism obtains profit $0$.

Taking expectation, this adaptive online mechanism obtains expected profit

$$(1 - \alpha)\left(\frac{\alpha}{c}\right)^{\frac{\alpha}{1-\alpha}} H^{\frac{1}{1-\alpha}} \left(1 - \left(1 - \frac{1}{nH}\right)^n\right) \geq (1 - \alpha)\left(\frac{\alpha}{c}\right)^{\frac{\alpha}{1-\alpha}} H^{\frac{\alpha}{1-\alpha}},$$

which is a lower bound of the optimal online mechanism. That is,

$$\mathrm{ONL} \geq (1 - \alpha)\left(\frac{\alpha}{c}\right)^{\frac{\alpha}{1-\alpha}} H^{\frac{\alpha}{1-\alpha}}.$$

**Approximation gap.** Therefore, for any non-adaptive mechanism,

$$\frac{\mathrm{ALG}}{\mathrm{ONL}} \leq H^{-\frac{\alpha}{1-\alpha}},$$

which tends to $0$ when $H$ is sufficiently large. This completes the proof. □

### B.4. Pseudocode for Two-stage Online Mechanism

---

**Algorithm 1** Two-stage Online Mechanism

---

1: **Input:** distributions $F_1, \ldots, F_n$, cost $c$, elasticity $\alpha \in (0, 1)$, parameter $\delta \in (0, 1)$
2: Compute $M = \sum_{i=1}^{n} \mathbb{E}[r_i]$
3: Set $D_0 = \left(\frac{\alpha}{c}(1 - \delta)M\right)^{1/(1-\alpha)}$
4: **for** $i = 1$ to $n$ **do**
5:     Observe reported value $\hat{v}_i$
6:     Compute $r_i \leftarrow \max\{\phi_i(\hat{v}_i), 0\}$
7:     **if** $r_i > 0$ **then**
8:         Set $x_i^{\mathrm{MG}} \leftarrow \left(\frac{\alpha r_i}{c}\right)^{1/(1-\alpha)}$
9:         Set $D_i \leftarrow \max\{D_{i-1}, x_i^{\mathrm{MG}}\}$ and allocate $x_i \leftarrow D_i$
10:        Charge payment $p_i$ according to Myerson's payment identity
11:     **else**
12:         Set $x_i \leftarrow 0$, $D_i \leftarrow D_{i-1}$
13:     **end if**
14: **end for**

---

## B.5. Proof of Theorem 3.4

*Proof.* We utilize Rosenthal's inequality (Lemma 3.5) to prove theorem 3.4.

To apply Lemma 3.5, we derive the offline optimal profit, denoted by OFFL, and derive a lower bound on the expected profit of Algorithm 1 With parameter $\delta = \frac{1-\alpha}{2-\alpha}$, denoted by ALG, stated in the following two lemmas.

**Lemma B.1.** $\mathrm{OFFL} = (1 - \alpha)(\frac{\alpha}{c})^{\frac{\alpha}{1-\alpha}} \mathbb{E}[(\sum_{i \in [n]} r_i)^{\frac{1}{1-\alpha}}]$.

**Lemma B.2.** $\mathrm{ALG} \geq \frac{1-\alpha}{2-\alpha}(1 - \alpha)(\frac{\alpha}{c})^{\frac{\alpha}{1-\alpha}} \cdot \max\{\sum_{i \in [n]} \mathbb{E}\left[r_i^{\frac{1}{1-\alpha}}\right], (\sum_{i \in [n]} \mathbb{E}[r_i])^{\frac{1}{1-\alpha}}\}$.

Applying Lemma 3.5, we have $\mathbb{E}[(\sum_{i \in [n]} r_i)^{\frac{1}{1-\alpha}}] \leq C_{\frac{1}{1-\alpha}} \cdot \max\{\sum_{i \in [n]} \mathbb{E}\left[r_i^{\frac{1}{1-\alpha}}\right], (\sum_{i \in [n]} \mathbb{E}[r_i])^{\frac{1}{1-\alpha}}\}$ holds with constant $C_{\frac{1}{1-\alpha}}$. Connecting with Lemma B.1 and Lemma B.2, we have $\mathrm{OFFL} \leq \frac{2-\alpha}{1-\alpha} C_{\frac{1}{1-\alpha}} \cdot \mathrm{ALG}$, which implies the theorem.

Below we prove Lemma B.1 and Lemma B.2.

For Lemma B.1, recall that the offline profit-optimal mechanism produces a data amount of

$$D^*(v_1, \cdots, v_n) = \arg\max_{x \geq 0} \sum_{i=1}^{n} \max\{\phi_i(v_i)x^\alpha, 0\} - cx$$

$$= \left(\frac{\alpha}{c} \sum_{i=1}^{n} r_i\right)^{\frac{1}{1-\alpha}}$$

And its expected profit is

$$\mathbb{E}[\sum_{i=1}^{n} r_i \cdot D^*(v_1, \cdots, v_n)^\alpha - cD^*(v_1, \cdots, v_n)]$$

$$= \mathbb{E}[(1 - \alpha)(\frac{\alpha}{c})^{\frac{\alpha}{1-\alpha}} (\sum_{i=1}^{n} r_i)^{\frac{1}{1-\alpha}}]$$

$$= (1 - \alpha)(\frac{\alpha}{c})^{\frac{\alpha}{1-\alpha}} \mathbb{E}[(\sum_{i=1}^{n} r_i)^{\frac{1}{1-\alpha}}].$$

This proves Lemma B.1.

Next, we prove Lemma B.2. Let $T = (1 - \delta)M$, recall that $D_0 = (\frac{T}{c})^{\frac{1}{1-\alpha}}$. Recall that the two-stage online mechanism (Algorithm 1) produces data in the online stage only when there is some $r_i > T$. More specifically, the amount of data at each round is $D_i = (\frac{\alpha}{c} \max\{T, r_1, \cdots, r_i\})^{\frac{1}{1-\alpha}}$. Therefore, we have

$$
\begin{aligned}
\text{ALG} &= \mathbb{E}[\sum_{i=1}^{n} r_i x_i^{\alpha} - cD_n] \\
&= \mathbb{E}[\sum_{i=1}^{n} r_i D_i^{\alpha} - cD_n] \\
&\geq (\frac{\alpha}{c})^{\frac{\alpha}{1-\alpha}} \mathbb{E}\Big[\sum_{i=1}^{n} r_i \max\{T, r_i\}^{\frac{\alpha}{1-\alpha}} \\
&\quad - \alpha \max\{T, r_1, \cdots, r_n\}^{\frac{1}{1-\alpha}}\Big]
\end{aligned}
\tag{5}
$$

Here the second equation is because $x_i = D_i$ whenever $r_i > 0$, and the inequality is because $D_i \geq (\frac{\alpha}{c} \max\{T, r_i\})^{\frac{1}{1-\alpha}}$ and $cD_n = c(\frac{\alpha}{c} \max\{T, r_1, \cdots, r_n\})^{\frac{1}{1-\alpha}} = \alpha(\frac{\alpha}{c})^{\frac{\alpha}{1-\alpha}} \max\{T, r_1, \cdots, r_n\}^{\frac{1}{1-\alpha}}$.

To lower bound ALG, we discuss two cases depending on the maximum of $r_1, \cdots, r_n$.

Case A. When $\max_{i \in [n]} r_i > T$, we can lower-bound the virtual welfare part,

$$
\begin{aligned}
&\sum_{i=1}^{n} r_i \max\{T, r_i\}^{\frac{\alpha}{1-\alpha}} \\
&= \sum_{i=1}^{n} r_i (T^{\frac{\alpha}{1-\alpha}} \mathbb{I}[r_i \leq T] + r_i^{\frac{\alpha}{1-\alpha}} \mathbb{I}[r_i > T]) \\
&\geq \sum_{i=1}^{n} r_i \mathbb{I}[r_i \leq T]((1-\delta)T^{\frac{\alpha}{1-\alpha}} + \delta r_i^{\frac{\alpha}{1-\alpha}}) + \sum_{i=1}^{n} \mathbb{I}[r_i > T] r_i^{\frac{1}{1-\alpha}}.
\end{aligned}
$$

And we can also lower-bound the cost part,

$$
\begin{aligned}
&- \alpha \max\{T, r_1, \cdots, r_n\}^{\frac{1}{1-\alpha}} \\
&= - (1-\delta)\alpha T^{\frac{1}{1-\alpha}} - \alpha \max_{i \in [n]}(r_i^{\frac{1}{1-\alpha}} - (1-\delta)T^{\frac{1}{1-\alpha}}) \\
&\geq - (1-\delta)\alpha T^{\frac{1}{1-\alpha}} - \alpha \sum_{i=1}^{n} \mathbb{I}[r_i > T](r_i^{\frac{1}{1-\alpha}} - (1-\delta)T^{\frac{1}{1-\alpha}}).
\end{aligned}
$$

Combining together, we have

$$
\begin{aligned}
&\sum_{i=1}^{n} r_i \max\{T, r_i\}^{\frac{\alpha}{1-\alpha}} - \alpha \max\{T, r_1, \cdots, r_n\}^{\frac{1}{1-\alpha}} \\
&\geq \sum_{i=1}^{n} r_i \mathbb{I}[r_i \leq T]((1-\delta)T^{\frac{\alpha}{1-\alpha}} + \delta r_i^{\frac{\alpha}{1-\alpha}}) - (1-\delta)\alpha T^{\frac{1}{1-\alpha}} \\
&\quad + \sum_{i=1}^{n} \mathbb{I}[r_i > T](r_i^{\frac{1}{1-\alpha}} - \alpha r_i^{\frac{1}{1-\alpha}} + (1-\delta)\alpha T^{\frac{1}{1-\alpha}})
\end{aligned}
\tag{6}
$$

Observe that $(1-\alpha)r_i^{\frac{1}{1-\alpha}} = \max_{x\geq 0} r_i x^\alpha - \alpha x \geq r_i(T^{\frac{1}{1-\alpha}})^\alpha - \alpha(T^{\frac{1}{1-\alpha}})$ for all $i \in [n]$, which implies

$$\sum_{i=1}^{n} \mathbb{I}[r_i > T](r_i^{\frac{1}{1-\alpha}} - \alpha r_i^{\frac{1}{1-\alpha}} + (1-\delta)\alpha T^{\frac{1}{1-\alpha}})$$

$$=\delta \sum_{i=1}^{n} \mathbb{I}[r_i > T](1-\alpha)r_i^{\frac{1}{1-\alpha}}$$

$$+ (1-\delta) \sum_{i=1}^{n} \mathbb{I}[r_i > T]((1-\alpha)r_i^{\frac{1}{1-\alpha}} + \alpha T^{\frac{1}{1-\alpha}})$$

$$\geq \delta \sum_{i=1}^{n} \mathbb{I}[r_i > T](1-\alpha)r_i^{\frac{1}{1-\alpha}} + (1-\delta) \sum_{i=1}^{n} \mathbb{I}[r_i > T]r_i T^{\frac{\alpha}{1-\alpha}}$$

Combining with (6), we have

$$\sum_{i=1}^{n} r_i \max\{T, r_i\}^{\frac{\alpha}{1-\alpha}} - \alpha \max\{T, r_1, \cdots, r_n\}^{\frac{1}{1-\alpha}}$$

$$\geq \sum_{i=1}^{n} r_i \mathbb{I}[r_i \leq T]((1-\delta)T^{\frac{\alpha}{1-\alpha}} + \delta r_i^{\frac{\alpha}{1-\alpha}})$$

$$- (1-\delta)\alpha T^{\frac{1}{1-\alpha}}$$

$$+ \delta \sum_{i=1}^{n} \mathbb{I}[r_i > T](1-\alpha)r_i^{\frac{1}{1-\alpha}}$$

$$+ (1-\delta) \sum_{i=1}^{n} \mathbb{I}[r_i > T]r_i T^{\frac{\alpha}{1-\alpha}}$$

$$=(1-\delta)(\sum_{i=1}^{n} r_i T^{\frac{\alpha}{1-\alpha}} - \alpha T^{\frac{1}{1-\alpha}})$$

$$+ \delta \sum_{i=1}^{n} (1 - \alpha \mathbb{I}[r_i > T])r_i^{\frac{1}{1-\alpha}}$$

$$\geq(1-\delta) \sum_{i=1}^{n} r_i T^{\frac{\alpha}{1-\alpha}} - \alpha T^{\frac{1}{1-\alpha}} + \delta \sum_{i=1}^{n} (1-\alpha)r_i^{\frac{1}{1-\alpha}}.$$

Here the inequality is because $(1-\delta)\alpha T^{\frac{1}{1-\alpha}} \leq \alpha T^{\frac{1}{1-\alpha}}$ and $1 - \alpha \mathbb{I}[r_i > T] \geq 1 - \alpha$.

Case B. When $\max_{i\in[n]} r_i \leq T$, we have

$$\sum_{i=1}^{n} r_i \max\{T, r_i\}^{\frac{\alpha}{1-\alpha}} - \alpha \max\{T, r_1, \cdots, r_n\}^{\frac{1}{1-\alpha}}$$

$$=\sum_{i=1}^{n} r_i T^{\frac{\alpha}{1-\alpha}} - \alpha T^{\frac{1}{1-\alpha}}$$

$$=(1-\delta) \sum_{i=1}^{n} r_i T^{\frac{\alpha}{1-\alpha}} - \alpha T^{\frac{1}{1-\alpha}} + \delta \sum_{i=1}^{n} r_i T^{\frac{\alpha}{1-\alpha}}$$

$$\geq(1-\delta) \sum_{i=1}^{n} r_i T^{\frac{\alpha}{1-\alpha}} - \alpha T^{\frac{1}{1-\alpha}} + \delta \sum_{i=1}^{n} (1-\alpha)r_i^{\frac{1}{1-\alpha}},$$

where the inequality is because $1 - \alpha < 1$ and $r_i \leq T$.

Combining cases A and B, it always holds that

$$\sum_{i=1}^{n} r_i \max\{T, r_i\}^{\frac{\alpha}{1-\alpha}} - \alpha \max\{T, r_1, \cdots, r_n\}^{\frac{1}{1-\alpha}}$$

$$\geq (1-\delta) \sum_{i=1}^{n} r_i T^{\frac{\alpha}{1-\alpha}} - \alpha T^{\frac{1}{1-\alpha}} + \delta(1-\alpha) \sum_{i=1}^{n} r_i^{\frac{1}{1-\alpha}}.$$

Taking expectation, recalling $T = (1-\delta)M = (1-\delta)\mathbb{E}\left[\sum_{i=1}^{n} r_i\right]$, we have

$$\mathbb{E}\left[\sum_{i=1}^{n} r_i \max\{T, r_i\}^{\frac{\alpha}{1-\alpha}} - \alpha \max\{T, r_1, \cdots, r_n\}^{\frac{1}{1-\alpha}}\right]$$

$$\geq \mathbb{E}\left[(1-\delta)\sum_{i=1}^{n} r_i\right] T^{\frac{\alpha}{1-\alpha}} - \alpha T^{\frac{1}{1-\alpha}} + \delta(1-\alpha)\sum_{i=1}^{n} \mathbb{E}\left[r_i^{\frac{1}{1-\alpha}}\right]$$

$$= T^{\frac{1}{1-\alpha}} - \alpha T^{\frac{1}{1-\alpha}} + \delta(1-\alpha)\sum_{i=1}^{n} \mathbb{E}\left[r_i^{\frac{1}{1-\alpha}}\right]$$

$$= (1-\alpha)\mathbb{E}\left[(1-\delta)\sum_{i=1}^{n} r_i\right]^{\frac{1}{1-\alpha}} + \delta(1-\alpha)\sum_{i=1}^{n} \mathbb{E}\left[r_i^{\frac{1}{1-\alpha}}\right].$$

It follows that

$$\mathbb{E}\left[\sum_{i=1}^{n} r_i \max\{T, r_i\}^{\frac{\alpha}{1-\alpha}} - \alpha \max\{T, r_1, \cdots, r_n\}^{\frac{1}{1-\alpha}}\right]$$

$$\geq (1-\alpha)(1-\delta)^{\frac{1}{1-\alpha}}\left(\sum_{i=1}^{n} \mathbb{E}[r_i]\right)^{\frac{1}{1-\alpha}} + \delta(1-\alpha)\sum_{i=1}^{n} \mathbb{E}\left[r_i^{\frac{1}{1-\alpha}}\right]$$

$$\geq (1-\alpha)\min\{(1-\delta)^{\frac{1}{1-\alpha}}, \delta\}$$

$$\cdot \max\left\{\left(\sum_{i=1}^{n} \mathbb{E}[r_i]\right)^{\frac{1}{1-\alpha}}, \sum_{i=1}^{n} \mathbb{E}\left[r_i^{\frac{1}{1-\alpha}}\right]\right\}.$$

We have $(1-\delta)^{\frac{1}{1-\alpha}} \geq 1 - \frac{\delta}{1-\alpha}$. Taking $\delta = \frac{1-\alpha}{2-\alpha}$, we have $\min\{(1-\delta)^{\frac{1}{1-\alpha}}, \delta\} = \frac{1-\alpha}{2-\alpha}$.

Combining with (5), we have

$$\text{ALG} \geq \left(\frac{\alpha}{c}\right)^{\frac{\alpha}{1-\alpha}}(1-\alpha)\frac{1-\alpha}{2-\alpha}\max\left\{\left(\sum_{i=1}^{n} \mathbb{E}[r_i]\right)^{\frac{1}{1-\alpha}}, \sum_{i=1}^{n} \mathbb{E}\left[r_i^{\frac{1}{1-\alpha}}\right]\right\},$$

which completes the proof of Lemma B.2 and Theorem 3.4. $\qquad\square$

### B.6. Proof of Theorem 3.6

*Proof.* For convenience, we assume $c = \alpha$ without loss of generality (for general $c > 0$, it is equivalent after scaling all values with a multiplicative factor $\frac{c}{\alpha}$).

Suppose $n$ is sufficiently large. We construct an instance $I_n$ with $n$ buyers, such that each buyer's virtual value $\phi_i(v_i)$ independently follows the Bernoulli distribution $\text{Bernoulli}(\frac{1}{n})$, i.e., $\phi_i(v_i) = 1$ with probability $\frac{1}{n}$. Specifically, each

buyer's value $v_i$ follows the truncated equal-revenue distribution with CDF $F_i(v_i) = \begin{cases} 1, & v_i \geq 1 \\ \max\{0, 1 - \frac{1}{nv_i}\}, & v_i \in [0, 1) \end{cases}$.

Our analysis consists of three steps. Firstly, we derive the expected profit $\text{ONL}(I_n)$ of the optimal online mechanism characterized by Proposition 3.1, and also calculate the expected profit $\text{OFFL}(I_n)$ of the optimal offline mechanism.

Secondly, we let $n$ tend to infinity, and rewrite the limits of ONL and OFFL as integrals. Thirdly, we show that the limit of OFFL can be expressed with $C_{\frac{1}{1-\alpha}}$, and derive an upper bound of $2^{O(\frac{1}{1-\alpha})}$ on the limit of ONL.

First, we explicitly characterize the optimal online mechanism for instance $I_n$ solved from the backward induction process in Proposition 3.1.

Recall the recursive characterization from Proposition 3.1: For $i = n, \cdots, 1$,

$$\Gamma_i(L) = \mathop{\mathbb{E}}_{v_i \sim F_i} [\max_{D_i \geq L} \max\{0, \phi_i(v_i)\} D_i^\alpha + \Gamma_{i+1}(D_i)]$$

with $\Gamma_{n+1}(L) := -cL = -\alpha L$ (assuming $c = \alpha$).

Under instance $I_n$, for each $i \in [n]$ we have $\max\{0, \phi_i(v_i)\} \sim \text{Bernoulli}(\frac{1}{n})$, so we have

$$\Gamma_i(L) = \frac{1}{n}(\max_{D_i \geq L} D_i^\alpha + \Gamma_{i+1}(D_i)) + (1 - \frac{1}{n})(\max_{D_i \geq L} \Gamma_{i+1}(D_i))$$

$\text{ONL}(I_n) = \Gamma_1(0)$. We prove the following structural result:

**Lemma B.3.** *Under instance $I_n$, for each $i = n, \cdots, 1$, $\Gamma_i(L)$ is concave and decreasing in $L \in [0, +\infty)$. Moreover, there is a threshold $T_i := (1 + \frac{n-i}{n})^{\frac{1}{1-\alpha}}$, such that*

$$\Gamma_i(L) = \begin{cases} (1 - \frac{1}{n})\Gamma_{i+1}(L) + \frac{1}{n}(1 - \alpha)(1 + \frac{n-i}{n})^{\frac{1}{1-\alpha}}, & L \leq T_i, \\ \frac{n-i+1}{n}L^\alpha - \alpha L, & L > T_i. \end{cases}$$

*Specifically, the optimal online mechanism for $I_n$ only produces data at most once. It produces $T_i$ amount of data when it encounters the first buyer $i$ with $v_i = 1$, and produces nothing if all values are zero.*

We prove Lemma B.3 by induction. For $i = n$, we can directly calculate

$$\Gamma_n(L) = \frac{1}{n}(\max_{D_n \geq L} D_n^\alpha - \alpha D_n) - (1 - \frac{1}{n})\alpha L$$

$$= \begin{cases} -(1 - \frac{1}{n})\alpha L + \frac{1}{n}(1 - \alpha), & L \leq 1 \\ \frac{1}{n}L^\alpha - \alpha L, & L > 1 \end{cases},$$

which satisfies the lemma, where $T_n = 1$. For any $i < n$, assume for induction that $\Gamma_{i+1}(L)$ satisfies the lemma.

Firstly, since both $D_i^\alpha$ and $\Gamma_{i+1}(D_i)$ are concave in $D_i$, we have $D_i^\alpha + \Gamma_{i+1}(D_i)$ is concave in $D_i$, and it is maximized at some $D_i^* \in [0, +\infty)$. Then we have $\max_{D_i \geq L} D_i^\alpha + \Gamma_{i+1}(D_i) = \begin{cases} D_i^{*\alpha} + \Gamma_{i+1}(D_i^*), & L \leq D_i^* \\ L^\alpha + \Gamma_{i+1}(L), & L \geq D_i^* \end{cases}$, which is concave and decreasing in $L$. Therefore, $\Gamma_i(L) = \frac{1}{n}(\max_{D_i \geq L} D_i^\alpha + \Gamma_{i+1}(D_i)) + (1 - \frac{1}{n})\Gamma_{i+1}(L)$ is concave and decreasing in $L$.

Moreover, by the lemma statement $\Gamma_{i+1}(L)$ is increasing on $[0, T_{i+1}]$, so $D_i^\alpha + \Gamma_{i+1}(D_i)$ is strictly increasng in $D_i \in [0, T_{i+1}]$, which implies that $D_i^* \geq T_{i+1}$. By induction assumption, we have

$$D_i^* = \arg \max_{D_i \geq T_{i+1}} D_i^\alpha + \Gamma_{i+1}(D_i)$$

$$= \arg \max_{D_i \geq T_{i+1}} D_i^\alpha + \frac{n - (i+1) + 1}{n}D_i^\alpha - \alpha D_i$$

$$= \arg \max_{D_i \geq T_{i+1}} (1 + \frac{n-i}{n})D_i^\alpha - \alpha D_i$$

$$= (1 + \frac{n-i}{n})^{\frac{1}{1-\alpha}} = T_i$$

Then, for any $L \geq 0$, we have

$$
\max_{D_i \geq L} D_i^\alpha + \Gamma_{i+1}(D_i)
$$

$$
= \begin{cases}
(1 + \frac{n-i}{n})D_i^{*\alpha} - \alpha D_i^*, & L \leq D_i^*, \\
(1 + \frac{n-i}{n})L^\alpha - \alpha L, & L \geq D_i^*,
\end{cases}
$$

$$
= \begin{cases}
(1 - \alpha)(1 + \frac{n-i}{n})^{\frac{1}{1-\alpha}}, & L \leq T_i, \\
(1 + \frac{n-i}{n})L^\alpha - \alpha L, & L \geq T_i.
\end{cases}
$$

Recall that $\Gamma_i(L) = \frac{1}{n}(\max_{D_i \geq L} D_i^\alpha + \Gamma_{i+1}(D_i)) + (1 - \frac{1}{n})(\max_{D_i \geq L} \Gamma_{i+1}(D_i))$. Since $\Gamma_{i+1}(D_i)$ is decreasing, we have $\max_{D_i \geq L} \Gamma_{i+1}(D_i) = \Gamma_{i+1}(L)$, and combining with the equation above, we have

$$
\Gamma_i(L) = \begin{cases}
(1 - \frac{1}{n})\Gamma_{i+1}(L) + \frac{1}{n}(1 - \alpha)(1 + \frac{n-i}{n})^{\frac{1}{1-\alpha}}, & L \leq T_i, \\
(1 - \frac{1}{n})\Gamma_{i+1}(L) + \frac{1}{n}((1 + \frac{n-i}{n})L^\alpha - \alpha L), & L \geq T_i,
\end{cases}
$$

$$
= \begin{cases}
(1 - \frac{1}{n})\Gamma_{i+1}(L) + \frac{1}{n}(1 - \alpha)(1 + \frac{n-i}{n})^{\frac{1}{1-\alpha}}, & L \leq T_i, \\
(1 - \frac{1}{n})(\frac{n-i}{n}L^\alpha - \alpha L) + \frac{1}{n}((1 + \frac{n-i}{n})L^\alpha - \alpha L), & L \geq T_i,
\end{cases}
$$

$$
= \begin{cases}
(1 - \frac{1}{n})\Gamma_{i+1}(L) + \frac{1}{n}(1 - \alpha)(1 + \frac{n-i}{n})^{\frac{1}{1-\alpha}}, & L \leq T_i, \\
\frac{n-i+1}{n}L^\alpha - \alpha L, & L \geq T_i.
\end{cases}
$$

That is, $\Gamma_i(L)$ satisfies the lemma. By induction, Lemma B.3 holds for all $i \in [n]$.

By Lemma B.3, we obtain

$$
\mathrm{ONL}(I_n) = \Gamma_1(0) = \sum_{i=1}^n (1 - \frac{1}{n})^{i-1} \frac{1}{n}(1 - \alpha)(1 + \frac{n-i}{n})^{\frac{1}{1-\alpha}}. \tag{7}
$$

Next, we derive $\mathrm{OFFL}(I_n)$:

$$
\mathrm{OFFL}(I_n) = \mathop{\mathbb{E}}_{v_1,\cdots,v_n}[(1 - \alpha)(\sum_{i \in [n]} \phi_i(v_i))^{\frac{1}{1-\alpha}}] = (1 - \alpha) \mathop{\mathbb{E}}_{Z \sim \mathrm{Binomial}(n, \frac{1}{n})}[Z^{\frac{1}{1-\alpha}}]. \tag{8}
$$

Now we consider the limit case when $n$ tends to infinity. From (7), we have

$$
\lim_{n \to \infty} \mathrm{ONL}(I_n) = \lim_{n \to \infty} \sum_{i=1}^n (1 - \frac{1}{n})^{i-1} \frac{1}{n}(1 - \alpha)(1 + \frac{n-i}{n})^{\frac{1}{1-\alpha}}
$$

$$
= (1 - \alpha) \int_0^1 e^{-t}(2 - t)^{\frac{1}{1-\alpha}} dt.
$$

From (8), we have

$$
\lim_{n \to \infty} \mathrm{OFFL}(I_n) = \lim_{n \to \infty} (1 - \alpha) \mathop{\mathbb{E}}_{Z \sim \mathrm{Binomial}(n, \frac{1}{n})}[Z^{\frac{1}{1-\alpha}}]
$$

$$
= (1 - \alpha) \mathop{\mathbb{E}}_{Z \sim \mathrm{Poisson}(1)}[Z^{\frac{1}{1-\alpha}}]
$$

$$
= (1 - \alpha)C_{\frac{1}{1-\alpha}}.
$$

Finally, since $\int_0^1 e^{-t}(2 - t)^{\frac{1}{1-\alpha}} dt < \int_0^1 e^{-t} 2^{\frac{1}{1-\alpha}} dt = (1 - e^{-1})2^{\frac{1}{1-\alpha}} < 2^{\frac{1}{1-\alpha}}$, we have

$$
\lim_{n \to \infty} \frac{\mathrm{ONL}(I_n)}{\mathrm{OFFL}(I_n)} = \frac{\int_0^1 e^{-t}(2 - t)^{\frac{1}{1-\alpha}} dt}{C_{\frac{1}{1-\alpha}}} < \frac{2^{\frac{1}{1-\alpha}}}{C_{\frac{1}{1-\alpha}}}.
$$

Therefore, for sufficiently large $n$, we have $\frac{\mathrm{ONL}(I_n)}{\mathrm{OFFL}(I_n)} \leq \frac{2^{\frac{1}{1-\alpha}}}{C_{\frac{1}{1-\alpha}}}$, which completes the proof. $\qquad\square$

## C. Missing Proofs in Section 4

### C.1. Proof of Payment Identities

We derive the expected payment identities and virtual value representations in section 4.1.

**Buyer side.** Fix $c$ and view the buyer as a single-parameter agent with value $v$. Let $u_B(v, c)$ denote the buyer's utility when truthfully reporting $v$. Incentive compatibility and individual rationality implies the envelope condition

$$\frac{\partial}{\partial v} u_B(v, c) = x(v, c)^\alpha, \qquad u_B(0, c) = 0,$$

hence

$$u_B(v, c) = \int_0^v x(t, c)^\alpha \, dt.$$

Using $u_B(v, c) = v x(v, c)^\alpha - p^B(v, c)$, the payment rule is

$$p^B(v, c) = v x(v, c)^\alpha - \int_0^v x(t, c)^\alpha \, dt.$$

Taking expectation and applying Fubini's theorem,

$$\mathbb{E}_{v,c}[p^B(v, c)] = \mathbb{E}_{v,c}[v x(v, c)^\alpha] - \mathbb{E}_c\left[\int_0^\infty x(t, c)^\alpha (1 - F^B(t)) dt\right]$$

$$= \mathbb{E}_{v,c}\left[\left(v - \frac{1 - F^B(v)}{f^B(v)}\right) x(v, c)^\alpha\right].$$

Defining $\phi^B(v) = v - \frac{1 - F^B(v)}{f^B(v)}$ yields

$$\mathbb{E}_{v,c}[p^B(v, c)] = \mathbb{E}_{v,c}\left[\phi^B(v) \, x(v, c)^\alpha\right].$$

**Seller side.** Fix $v$ and treat the seller as a single-parameter agent with cost $c$. Let $u_S(c, v)$ denote the seller's utility when truthfully reporting $c$. Incentive compatibility and individual rationality implies

$$\frac{\partial}{\partial c} u_S(c, v) = -x(v, c), \qquad u_S(\infty, v) = 0,$$

so

$$u_S(c, v) = \int_c^\infty x(v, t) \, dt.$$

Since $u_S(c, v) = p^S(v, c) - c x(v, c)$, we have

$$p^S(v, c) = c x(v, c) + \int_c^\infty x(v, t) \, dt.$$

Taking expectation,

$$\mathbb{E}_{v,c}[p^S(v, c)] = \mathbb{E}_{v,c}[c x(v, c)] + \mathbb{E}_v\left[\int_0^\infty x(v, t) F^S(t) \, dt\right]$$

$$= \mathbb{E}_{v,c}\left[\left(c + \frac{F^S(c)}{f^S(c)}\right) x(v, c)\right].$$

Defining $\phi^S(c) = c + \frac{F^S(c)}{f^S(c)}$ gives

$$\mathbb{E}_{v,c}[p^S(v, c)] = \mathbb{E}_{v,c}\left[\phi^S(c) \, x(v, c)\right].$$

## C.2. Derivation of Second Best Mechanism

Using the virtual value and virtual cost functions, the weak budget-balance constraint can be rewritten as:

$$\mathbb{E}_{v,c}\left[\phi^B(v)\cdot x(v,c)^\alpha - \phi^S(c)\cdot x(v,c)\right] \geq 0.$$

where $\phi^B(v)$ is the buyer's virtual value and $\phi^S(c)$ is the seller's virtual cost. We characterize the second-best mechanism via a Lagrangian relaxation of the budget balance constraint. Let $\lambda \geq 0$ denote the Lagrange multiplier associated with this constraint. The resulting relaxed problem can be written as:

$$\max_{x(\cdot,\cdot)} \mathbb{E}_{v,c}\left[(v + \lambda\phi^B(v))\cdot x(v,c)^\alpha - (c + \lambda\phi^S(c))\cdot x(v,c)\right]. \tag{9}$$

The unique maximizer for equation 9 is given by

$$x(v,c) = \left(\frac{\alpha(v + \lambda\phi^B(v))}{c + \lambda\phi^S(c)}\right)^{\frac{1}{1-\alpha}}$$

whenever the numerator and denominator are positive, and $x(v,c) = 0$ otherwise. This yields the allocation rule of second-best mechanism stated in Section 4.2. When $F^B$ is regular, since $\phi^B(v)$ is increasing in $v$ and $\lambda \geq 0$, the numerator is increasing in $v$. Similarly, when $F^S$ is regular (for virtual cost), $\phi^S(c)$ is increasing in $c$, and the denominator is increasing in $c$. For irregular distributions, the virtual value or virtual cost function is replaced by the ironed version to ensure monotonicity.

By construction, the allocation is monotone increasing in $v$ and decreasing in $c$, hence implementable by a truthful and individually rational mechanism. The payments follow from Myerson's payment identities.

Finally, since the budget balance constraint is linear and the relaxed problem is concave, strong duality holds. Therefore, there exists a multiplier $\lambda^* \geq 0$ such that the budget balance constraint holds with equality, which completes the characterization of the second-best mechanism.

## C.3. Preparations for Theorem 4.1 and Theorem 4.2

Before proving Theorem 4.1 and Theorem 4.2, we formally state the welfare decomposition technique in the following technical lemma.

**Lemma C.1.** *Given $\alpha \in (0,1)$, define function*

$$W(v,c) := \max_{x \geq 0} v \cdot x^\alpha - cx$$

*for $v \in (-\infty, +\infty)$ and $c \in (0, +\infty)$, i.e., the first-best GFT when buyer has value $v$ and seller has cost c. Here we allow $v$ to be negative for technical convenience. Then we have*

$$W(v,c) = (1-\alpha)\alpha^{\frac{\alpha}{1-\alpha}} \cdot (v)_+^{\frac{1}{1-\alpha}} \cdot c^{-\frac{\alpha}{1-\alpha}},$$

*which is achieved by $x = (\frac{\alpha v}{c})_+^{\frac{1}{1-\alpha}}$.*

*Specifically, define $W^B(v) := (v)_+^{\frac{1}{1-\alpha}}$ and $W^S(c) := c^{-\frac{\alpha}{1-\alpha}}$, we can write $W(v,c)$ as*

$$W(v,c) = (1-\alpha)\alpha^{\frac{\alpha}{1-\alpha}} \cdot W^B(v) \cdot W^S(c).$$

*Then the first-best gain-from-trade can be expressed as*

$$\text{FB} = (1-\alpha)\alpha^{\frac{\alpha}{1-\alpha}} \cdot \mathbb{E}_{v \sim F^B}[W^B(v)] \cdot \mathbb{E}_{c \sim F^S}[W^S(c)].$$

*Proof.* For $v \leq 0$, we have $\max_{x \geq 0} v \cdot x^\alpha - cx \leq 0$, so the maximum is $W(v,c) = 0$, obtained with $x = 0$. For $v > 0$, by solving $\max_{x \geq 0} v \cdot x^\alpha - cx$ with first order condition, the maximum is obtained at $x = (\frac{\alpha v}{c})^{\frac{1}{1-\alpha}}$. Combining the two cases,

the optimal solution is $x = (\frac{\alpha v}{c})_+^{\frac{1}{1-\alpha}}$. It follows

$$W(v,c) = (1-\alpha)\alpha^{\frac{\alpha}{1-\alpha}} \cdot (v)_+^{\frac{1}{1-\alpha}} \cdot c^{-\frac{\alpha}{1-\alpha}}$$
$$= (1-\alpha)\alpha^{\frac{\alpha}{1-\alpha}} \cdot W^B(v) \cdot W^S(c).$$

By definition of FB, we have

$$\text{FB} = \underset{v\sim F^B, c\sim F^S}{\mathbb{E}}[W(v,c)]$$
$$= \underset{v\sim F^B, c\sim F^S}{\mathbb{E}}[1-\alpha)\alpha^{\frac{\alpha}{1-\alpha}} \cdot W^B(v) \cdot W^S(c)]$$
$$= (1-\alpha)\alpha^{\frac{\alpha}{1-\alpha}} \cdot \underset{v\sim F^B}{\mathbb{E}}[W^B(v)] \cdot \underset{c\sim F^S}{\mathbb{E}}[W^S(c)].$$

The last equation is because $v, c$ are independent. $\qquad\square$

We will also apply the following Hardy's inequality.

**Lemma C.2** (Hardy (1920)). *Let $f(x)$ be a non-negative integrable function on $[0,a]$. Define $F(x) = \int_0^x f(t)dt$. Then for any $p < 0$ or $p > 1$,*

$$\int_0^a (\frac{F(x)}{x})^p dx \leq (\frac{p}{p-1})^p \int_0^a (f(x))^p dx.$$

### C.4. Proof of Theorem 4.1

*Proof.* In this proof, we firstly utilize the welfare decomposition technique to derive a lower bound on the approximation ratio, which only depends on the buyer's value distribution $F^B$. The relationship between the quantile-space representation of value distribution and virtual value distribution then allows us to apply Hardy's inequality and derive a constant bound.

For convenience, we assume $F^B$ is regular for now, and we will show that this is without loss of generality later.

Firstly, since the gain-from-trade SellerP is at least seller's expected utility, which can be expressed as seller's virtual surplus, we have

$$\text{SellerP} \geq \text{SellerU} := \underset{v\sim F^B, c\sim F^S}{\mathbb{E}}[\max_{x\geq 0} \phi^B(v)x^\alpha - cx]$$
$$= \underset{v\sim F^B, c\sim F^S}{\mathbb{E}}[W(\phi^B(v), c)]$$
$$= \underset{v\sim F^B, c\sim F^S}{\mathbb{E}}[(1-\alpha)\alpha^{\frac{\alpha}{1-\alpha}} \cdot W^B(\phi^B(v)) \cdot W^S(c)]$$
$$= (1-\alpha)\alpha^{\frac{\alpha}{1-\alpha}} \underset{v\sim F^B}{\mathbb{E}}[W^B(\phi^B(v))] \underset{c\sim F^S}{\mathbb{E}}[W^S(c)].$$

Here the third equation is by Lemma C.1, the fourth equation is by the independence of $v$ and $c$.

By Lemma C.1, similarly, we express the first-best gain-from-trade as

$$\text{FB} = (1-\alpha)\alpha^{\frac{\alpha}{1-\alpha}} \underset{v\sim F^B}{\mathbb{E}}[W^B(v)] \underset{c\sim F^S}{\mathbb{E}}[W^S(c)].$$

Combining together, we obtain

$$\frac{\text{SellerU}}{\text{FB}} = \frac{\mathbb{E}_{v\sim F^B}[W^B(\phi^B(v))]}{\mathbb{E}_{v\sim F^B}[W^B(v)]}.$$

Therefore, we aim to prove that

$$\inf_{F^B} \frac{\mathbb{E}_{v\sim F^B}[W^B(\phi^B(v))]}{\mathbb{E}_{v\sim F^B}[W^B(v)]} \geq \alpha^{\frac{1}{1-\alpha}}.$$

For convenience, we express a distribution $F^B$ in quantile space. Define quantile function $v^B(q) = (F^B)^{-1}(1-q)$. $v^B(q)$ is weakly decreasing in $q \in [0,1]$. Viewing $q$ as a random variable, it is well-known that the distribution of $v^B(q)$ is $F^B$ when $q$ is drawn from $U[0,1]$. We can rewrite

$$\frac{\mathbb{E}_{v \sim F^B}[W^B(\phi^B(v))]}{\mathbb{E}_{v \sim F^B}[W^B(v)]} = \frac{\mathbb{E}_{q \sim U[0,1]}[W^B(\phi^B(v^B(q)))]}{\mathbb{E}_{q \sim U[0,1]}[W^B(v^B(q))]}.$$

Define the revenue curve $R^B(q) = q \cdot v^B(q)$, it holds that $(R^B)'(q) = \phi^B(v^B(q))$. It follows that $v^B(q) = \frac{R^B(q)}{q} = \frac{1}{q}\left(\int_0^q \phi^B(v^B(t))dt\right)$. For convenience define the virtual value in quantile space $\psi(q) = \phi^B(v^B(q))$, a distribution $F^B$ is regular if and only if $\psi(q)$ is non-increasing in $q \in [0,1]$.

Now we show that the assumption that $F^B$ is regular is without loss of generality. Suppose $F^B$ is an irregular distribution, then we replace $\phi^B(v)$ with the ironed virtual value $\phi_{ir}^B(v)$ in the above derivation, and it leads to a ratio

$$\frac{\text{SellerU}}{\text{FB}} = \frac{\mathbb{E}_{v \sim F^B}[W^B(\phi_{ir}^B(v))]}{\mathbb{E}_{v \sim F^B}[W^B(v)]}.$$

Now consider a regular distribution such that its virtual value in quantile space is $\psi_+(q) = \phi_{ir}^B(v^B(q))$. Denote this distribution by $F_+^B$, and define $v_+^B(q)$ denoting its corresponding quantile function, and $R_+^B(q)$ denoting its revenue curve. By (Myerson, 1981), $R_+^B(q)$ is the upper convex hull of $R^B(q)$. Therefore, $v_+^B(q) = \frac{R_+^B(q)}{q} \geq \frac{R^B(q)}{q} = v^B(q)$. Then we have

$$\frac{\mathbb{E}_{v \sim F^B}[W^B(\phi_{ir}^B(v))]}{\mathbb{E}_{v \sim F^B}[W^B(v)]} = \frac{\mathbb{E}_{q \sim U[0,1]}[W^B(\phi_{ir}^B(v^B(q)))]}{\mathbb{E}_{q \sim U[0,1]}[W^B(v^B(q))]} \geq \frac{\mathbb{E}_{q \sim U[0,1]}[W^B(\psi_+(q))]}{\mathbb{E}_{q \sim U[0,1]}[W^B(v_+^B(q))]} = \frac{\mathbb{E}_{v \sim F_+^B}[W^B(\phi_+^B(v))]}{\mathbb{E}_{v \sim F_+^B}[W^B(v)]},$$

where the inequality is because $\phi_{ir}^B(v^B(q)) = \psi_+(q)$ but $v^B(q) \leq v_+^B(q)$, and $W^B$ is non-decreasing. That is, there is a regular distribution $F_+^B$ leading to a (weakly) worse ratio than the irregular distribution $F^B$, so it is without loss of generality to only consider regular distributions in our worst-case analysis.

Similarly, we can also assume the virtual values to be non-negative, i.e. $\psi(q) \geq 0$. The reason is, supposing that $\psi(q) < 0$ for some $q$, we can define $\psi_+(q) = \max\{\psi(q), 0\}$ and consider the value distribution $F_+^B$ given by $v_+^B(q) = \frac{1}{q}\left(\int_0^q \psi_+(t)dt\right)$. We have $v_+^B(q) = \frac{1}{q}\left(\int_0^q \psi_+(t)dt\right) \geq \frac{1}{q}\left(\int_0^q \psi(t)dt\right) = v^B(q)$. It follows that $\mathbb{E}_{q \sim U[0,1]}[W^B(\psi(q))dq] = \mathbb{E}_{q \sim U[0,1]}[W^B(\psi_+(t))]$ but $\mathbb{E}_{q \sim U[0,1]}[W^B(v^B(q))] \leq \mathbb{E}_{q \sim U[0,1]}[W^B(v_+^B(q))]$, that is, the constructed distribution with non-negative virtual values is weakly worse.

Now we analyze the worst case, focusing on regular distributions with non-negative virtual values. We rewrite

$$\frac{\text{SellerU}}{\text{FB}} = \frac{\mathbb{E}_{q \sim U[0,1]}[W^B(\phi^B(v^B(q)))]}{\mathbb{E}_{q \sim U[0,1]}[W^B(v^B(q))]}$$

$$= \frac{\int_0^1 W^B(\psi(q))dq}{\int_0^1 W^B(\frac{1}{q}\int_0^q \psi(t)dt)dq}.$$

Recall that $W^B(v) = \max\{v, 0\}^{\frac{1}{1-\alpha}}$. We have

$$\frac{\int_0^1 W^B(\psi(q))dq}{\int_0^1 W^B(\frac{1}{q}\int_0^q \psi(t)dt)dq} = \frac{\int_0^1 (\psi(q))^{\frac{1}{1-\alpha}}dq}{\int_0^1 (\frac{1}{q}\int_0^q \psi(t)dt)^{\frac{1}{1-\alpha}}dq}.$$

By Hardy's inequality (Lemma C.2), we have

$$\int_0^1 \left(\frac{1}{q}\int_0^q \psi(t)dt\right)^{\frac{1}{1-\alpha}}dq \leq \left(\frac{\frac{1}{1-\alpha}}{\frac{1}{1-\alpha}-1}\right)^{\frac{1}{1-\alpha}}\int_0^1 (\psi(q))^{\frac{1}{1-\alpha}}dq$$

$$= \left(\frac{1}{\alpha}\right)^{\frac{1}{1-\alpha}}\int_0^1 (\psi(q))^{\frac{1}{1-\alpha}}dq.$$

It follows that

$$\inf_{F^B} \frac{\mathbb{E}_{v\sim F^B}[W^B(\phi^B(v))]}{\mathbb{E}_{v\sim F^B}[W^B(v)]}$$

$$= \inf_{\psi(\cdot)} \frac{\int_0^1 (\psi(q))^{\frac{1}{1-\alpha}} dq}{\int_0^1 (\frac{1}{q} \int_0^q \psi(t)dt)^{\frac{1}{1-\alpha}} dq}$$

$$\geq \alpha^{\frac{1}{1-\alpha}}.$$

That is, $\frac{\mathrm{SellerP}}{\mathrm{FB}} \geq \frac{\mathrm{SellerU}}{\mathrm{FB}} \geq \alpha^{\frac{1}{1-\alpha}}$ for any distribution $F^B$. This completes the proof of Theorem 4.1. $\square$

## C.5. Proof of Theorem 4.2

*Proof.* This proof follows similar steps as the proof of Theorem 4.1. We firstly utilize the welfare decomposition technique to derive a lower bound on the approximation ratio, which only depends on the seller's cost distribution $F^S$. The relationship between the quantile-space representation of cost distribution and virtual cost distribution then allows us to apply Hardy's inequality and derive a constant bound.

For convenience, we assume the distribution $F^S$ is regular for virtual cost, i.e., $\phi^S(c)$ is non-decreasing in $c$. Later we will show that this is without loss of generality.

Similar to the proof of Theorem 4.1, we can lower-bound the gain-from-trade $\mathrm{BuyerP}$ by buyer's utility in buyer-proposing mechanism, equal to the virtual surplus

$$\mathrm{BuyerP} \geq \mathrm{BuyerU} := \mathbb{E}_{v\sim F^B, c\sim F^S}[\max_{x\geq 0} vx^\alpha - \phi^S(c)x]$$

$$= \mathbb{E}_{v\sim F^B, c\sim F^S}[W(v, \phi^S(c))]$$

$$= (1-\alpha)\alpha^{\frac{\alpha}{1-\alpha}} \mathbb{E}_{v\sim F^B}[W^B(v)] \mathbb{E}_{c\sim F^S}[W^S(\phi^S(c))].$$

The second and third equalities are by Lemma C.1 and independence of $v$ and $c$.

By Lemma C.1, the first-best gain-from-trade can be expressed as

$$\mathrm{FB} = (1-\alpha)\alpha^{\frac{\alpha}{1-\alpha}} \mathbb{E}_{v\sim F^B}[W^B(v)] \mathbb{E}_{c\sim F^S}[W^S(c)].$$

Therefore, we have

$$\frac{\mathrm{BuyerU}}{\mathrm{FB}} = \frac{\mathbb{E}_{c\sim F^S}[W^S(\phi^S(c))]}{\mathbb{E}_{c\sim F^S}[W^S(c)]}.$$

And we aim to prove that

$$\inf_{F^S} \frac{\mathbb{E}_{c\sim F^S}[W^S(\phi^S(c))]}{\mathbb{E}_{c\sim F^S}[W^S(c)]} \geq \alpha^{\frac{\alpha}{1-\alpha}}.$$

Without loss of generality, assume the support of $F^S$ is non-negative and bounded. For convenience, we express the distribution $F^S$ in quantile space. Define quantile function $c^S(q) = (F^S)^{-1}(q)$. $c^S(q)$ is weakly increasing in $q \in [0, 1]$. Viewing $q$ as a random variable, it is known that the distribution of $c^S(q)$ is $F^S$ when $q$ is drawn from $U[0, 1]$.

Define the cost curve $R^S(q) = q \cdot c^S(q)$, it holds that $(R^S)'(q) = \phi^S(c^S(q))$. It follows that $c^S(q) = \frac{R^S(q)}{q} = \frac{1}{q}\left(\int_0^q \phi^S(c^S(t))dt\right)$.

Now we show that assuming $F^S$ is regular for virtual cost is without loss of generality, similar to the proof of Theorem 4.1. Suppose $F^S$ is not regular for virtual cost, i.e., $\phi^S(c^S(q))$ is not increasing in $q$. we replace $\phi^S(c)$ by $\phi_{\mathrm{ir}}^S(c)$ in the derivation above, which leads to a ratio $\frac{\mathrm{BuyerU}}{\mathrm{FB}} = \frac{\mathbb{E}_{c\sim F^S}[W^S(\phi_{\mathrm{ir}}^S(c))]}{\mathbb{E}_{c\sim F^S}[W^S(c)]}$. Define $F_-^S$ as the distribution with virtual cost in quantile space

$\psi_-(q) = \phi^S_{\text{ir}}(c(q))$. Its cost curve $R^S_-(q)$ is the lower convex hull of $R^S(q)$, and its quantile function is $c^S_-(q) = \frac{1}{q}R^S_-(q) \leq \frac{1}{q}R^S(q) = c(q)$. Since $W^S(c) = c^{-\frac{\alpha}{1-\alpha}}$ is decreasing in $c$, it holds that $W^S(c^S(q)) \leq W^S(c^S_-(q))$ for $q \in [0,1]$. Then we have $\mathbb{E}_{c\sim F^S}[W^S(\phi^S_{\text{ir}}(c))] = \mathbb{E}_{q\sim U[0,1]}[W^S(\phi^S_{\text{ir}}(c(q)))] = \mathbb{E}_{q\sim U[0,1]}[W^S(\psi_-(q))] = \mathbb{E}_{c\sim F^S_-}[W^S(\phi^S_{\text{ir}}(c))]$, but $\mathbb{E}_{c\sim F^S}[W^S(c)] = \mathbb{E}_{q\sim U[0,1]}[W^S(c^S(q))] \leq \mathbb{E}_{q\sim U[0,1]}[W^S(c^S_-(q))] = \mathbb{E}_{c\sim F^S_-}[W^S(c)]$. That is, $F^S_-$ is a weakly worse distribution that is regular for virtual cost. Therefore, we only need to consider regular distributions in worst-case analysis.

Now we analyze the worst case, focusing on regular distributions. For convenience define $\psi(q) = \phi^S(c^S(q))$, then $\psi(q)$ is non-decreasing in $q \in [0,1]$. We can rewrite

$$\frac{\mathbb{E}_{c\sim F^S}[W^S(\phi^S(c))]}{\mathbb{E}_{c\sim F^S}[W^S(c)]} = \frac{\mathbb{E}_{q\sim U[0,1]}[W^S(\phi^S(c^S(q)))]}{\mathbb{E}_{q\sim U[0,1]}[W^S(c^S(q))]}$$

$$= \frac{\int_0^1 W^S(\psi(q))dq}{\int_0^1 W^S(\frac{1}{q}\int_0^q \psi(t)dt)dq}$$

$$= \frac{\int_0^1 (\psi(q))^{-\frac{\alpha}{1-\alpha}}dq}{\int_0^1 (\frac{1}{q}\int_0^q \psi(t)dt)^{-\frac{\alpha}{1-\alpha}}dq}.$$

Here we recall that $W^S(c) = c^{-\frac{\alpha}{1-\alpha}}$.

We have that $\psi(q)$ is non-negative since $F^S$ is non-negative. Applying Lemma C.2 with $p = -\frac{\alpha}{1-\alpha} < 0$, we have

$$\frac{\int_0^1 (\frac{1}{q}\int_0^q \psi(t)dt)^{-\frac{\alpha}{1-\alpha}}dq}{\int_0^1 (\psi(q))^{-\frac{\alpha}{1-\alpha}}dq} \leq (\frac{-\frac{\alpha}{1-\alpha}}{-\frac{\alpha}{1-\alpha}-1})^{-\frac{\alpha}{1-\alpha}}$$

$$= \alpha^{-\frac{\alpha}{1-\alpha}}.$$

It follows that

$$\inf_{F^S} \frac{\mathbb{E}_{c\sim F^S}[W^S(\phi^S(c))]}{\mathbb{E}_{c\sim F^S}[W^S(c)]}$$

$$= \inf_{\psi(\cdot)} \frac{\int_0^1 (\psi(q))^{-\frac{\alpha}{1-\alpha}}dq}{\int_0^1 (\frac{1}{q}\int_0^q \psi(t)dt)^{-\frac{\alpha}{1-\alpha}}dq}$$

$$\geq \alpha^{\frac{\alpha}{1-\alpha}}.$$

That is, $\frac{\text{BuyerP}}{\text{FB}} \geq \frac{\text{BuyerU}}{\text{FB}} \geq \alpha^{\frac{\alpha}{1-\alpha}}$ for all distribution $F^S$.

Finally, since $\alpha = (1 + \frac{1-\alpha}{\alpha})^{-1}$ and $(1 + \frac{1-\alpha}{\alpha})^{\frac{\alpha}{1-\alpha}} \leq e$, we have $\alpha^{\frac{\alpha}{1-\alpha}} = \frac{1}{(1+\frac{1-\alpha}{\alpha})^{\frac{\alpha}{1-\alpha}}} \geq \frac{1}{e}$.

This completes the proof.

$\square$

