# OpenReview forum: "Selling Data as a Digital Good with Scaling Valuations"
_ICML.cc/2026/Conference — ICML 2026 regular_

### Official Review · Reviewer_NWYa · 2026-03-08

**Soundness:** 3
**Presentation:** 3
**Significance:** 3
**Originality:** 4
**Overall Recommendation:** 6
**Confidence:** 4

**Summary:**

The paper studies the problem of selling data as a digital good with following unique properties:

--- The same data can be sold to multiple buyers simultaneously;

--- For any buyer, the utility of getting data is diminishing with power law.

Particularly, the authors consider the following three settings:

--- Offline setting in which the seller wants to optimize his total revenue. In this setting, the authors proposes an auction-like mechanism with Myerson's lemma and optimizes revenue while ensuring incentives.

--- Online setting in which the buyer comes one by one and the seller must make immediate decisions to each buyer; In this setting, the authors show that naive greedy or non-adaptive methods may get arbitrarily bad revenue, and then provide a DP-based optimal solution as well as a simpler online mechanism that gives constant factor approximation of optimal revenue. The authors then provide a online-offline revenue gap result with analysis on the second mechanism.

--- The meditator setting in which we act as a meditator instead of the seller and want to optimize the total gain-from-trade (i.e., social welfare). The authors then provide two mechanisms, seller-propose and buyer-propose, and show that both achieve constant factor approximation of first-best welfare.

**Compliance With Llm Reviewing Policy:**

Affirmed.

**Final Justification:**

The authors resolved my question, and I do believe that although some technical details are open to future study, this work’s modeling and novelty would have a good potential for those who follow up in the field of data markets.

One paper may not be able to settle every question in a field, but we need this kind of work that explores the boundaries for those who come after.

**Key Questions For Authors:**

--- If we want to allow different $\alpha$ for different buyers, is there any technical challenge that makes it harder?

--- In the meditator setting, do you have any ideas if the target is to maximize the utility of the seller, buyer, or meditator?

**Limitations:**

Yes.

**Strengths And Weaknesses:**

Strengths:

--- Novel and elegant modeling on the special property of data as a commodity, providing insight for the marketplace of data with distinction from other kind of goods that are mostly rivalrous.

--- Thorough and theoretically guaranteed analysis on different cases, providing solution for different scenario of the data market.

Weaknesses (minor):

--- Buyer valuation model is a little restrictive. Actually different buyers may have heterogeneous systems with different scalability w.r.t. data, i.e., different $\alpha$ (like "The Bitter Lesson" discuss). Considering different $\alpha$ (or even general scaling function) will give this study more general value.

---

> ### Author Rebuttal · Authors · 2026-03-30
>
> Thank you for your constructive feedback. Below is our response to your questions.
>
> **Q1 Extension to heterogeneous scaling exponents.**
> We acknowledge that extending our results to heterogenous $\alpha$'s is an important direction for future work, as discussed in Section 5. Actually, we believe the characterization of optimal online or bilateral mechanisms (in Section 3.2, 4.2) can be extended to heterogeneous scaling exponents within the current virtual surplus maximizing framework, analogous to the generalization for the offline setting in Appendix A.4. However, extending our main results concerning the approximation bounds faces significant technical challenges and likely requires different analytical approaches:
> - Firstly, under heterogeneous scaling exponents, the allocation and resulting revenue of optimal mechanisms are given implicitly by an optimization problem without closed-form solutions, so the approximation targets will no longer have clear formulas.
> - Secondly, deriving approximation ratios in online and bilateral settings will require significantly more involved techniques, as both Rosenthal's inequality (Lemma 3.5) and Hardy's inequality (Lemma C.2) are inapplicable without the power-law forms of revenues.
> - Thirdly, if the heterogeneous scaling parameters are private to each buyer, the mechanism design problem becomes multi-dimensional, where each buyer may misreport $v_i$ and $\alpha_i$ if not properly incentivized. Optimal mechanisms are typically intractable in such settings due to intricate incentive compatibility constraints.
>
> **Q2 Bilateral trading mechanisms maximizing utility of seller, buyer, or mediator.**
> The optimal mechanisms for all of these three objectives are tractable. In fact, the seller-proposing and buyer-proposing mechanisms (in Section 4.3) are exactly the optimal mechanisms for seller's and buyer's utility, respectively. For example, the allocation rule of the seller-proposing mechanism maximizes the seller's virtual surplus, so the seller's utility, equal to the expectation of virtual surplus, is also maximized. The optimal mechanism for the mediator's utility is specified by the allocation rule $x(v,c)=(\frac{\alpha\phi^B(v)}{\phi^S(c)})_+^{\frac1{1-\alpha}}$, calculated based on the virtual value/cost of both seller and buyer, interpreted as ironed versions when distributions are irregular. This maximizes the virtual surplus for the mediator, which is defined with both virtual value and virtual cost.

---

> > ### Author Rebuttal · Reviewer_NWYa · 2026-03-31
> >
> > The authors resolved my question. I am positive on this paper anyway.

---

> > > ### Author Response · Authors · 2026-04-04
> > >
> > > Thank you very much for your time and for your positive recognition of our contribution. We are glad that our response helped address your questions.

---

### Official Review · Reviewer_LSUZ · 2026-03-12

**Soundness:** 3
**Presentation:** 3
**Significance:** 3
**Originality:** 3
**Overall Recommendation:** 4
**Confidence:** 3

**Summary:**

The authors explore mechanism design for selling data as a digital good specifically when buyer (AI models, e.g.) valuations follow a scaling law. Unlike classical auction models with binary allocations, this framework allows for continuous quantities of data rather than binary outcomes.  Key contributions include -

1. Offline Profit Maximization: The authors characterize optimal mechanisms when all buyer types are known simultaneously, showing that virtual-value methods extend to continuous allocations.
2. Online Production: In sequential arrival settings where production is irreversible, their results show that myopic or fixed production plans are suboptimal. They propose a two-stage adaptive algorithm that achieves a constant-factor approximation to the offline optimum.
3. Bilateral Data Trading: For trade under asymmetric information (private values and private seller costs), the paper identifies simple mechanisms that recover a constant fraction of the first-best gain-from-trade.

**Compliance With Llm Reviewing Policy:**

Affirmed.

**Key Questions For Authors:**

How can your model handle truly heterogeneous buyers?  In particular, when different buyers have different utility for the data.

**Limitations:**

Yes

**Strengths And Weaknesses:**

Strengths:

1. The paper effectively bridges the gap between empirical ML observations (scaling laws) and formal economic theory. The use of multiplicative welfare decomposition is a technically elegant way to derive robust approximation ratios that are independent of the underlying distributions.

2. Highly significant for the growing field of data marketplaces (e.g., Snowflake, AWS Data Exchange) where data is a continuously produced commodity.

Weaknesses:

1. The analysis assumes the scaling parameter $\alpha$ is fixed and known to the seller. In practice, this parameter might be heterogeneous across buyers or unknown, which is left as an area for future work.

---

> ### Author Rebuttal · Authors · 2026-03-30
>
> Thank you for your valuable feedback. Below is our response to your comments.
>
> **Q1 & W1** We acknowledge that it is an important future direction of our work to extend the results to general and heterogeneous valuations. For example, a direct generalization is to consider heterogeneous values of $\alpha$, and a further generalization is to consider the general class of concave functions as valuations.
> Actually, we believe the characterization of optimal online or bilateral mechanisms (in Section 3.2, 4.2) can be extended to heterogeneous concave valuation functions within the current virtual surplus maximizing framework, analogous to the generalization for the offline setting in Appendix A.4. However, extending our main results concerning the approximation bounds will likely require substantially different techniques. For example:
> - When valuation functions have heterogeneous scaling behaviors (e.g., different $\alpha_i$ for each buyer), the allocation and resulting revenue of optimal mechanisms no longer have closed forms, so the approximation targets will no longer have clear formulas.
> - Deriving constant approximation ratios in online and bilateral settings would require significantly more involved techniques, as both Rosenthal's inequality (Lemma 3.5) and Hardy's inequality (Lemma C.2) are inapplicable without the power-law forms.
> - If the heterogeneous scaling parameters or the entire valuation functions are private to each buyer, the mechanism design problem becomes multi-dimensional or even infinite-dimensional, where the optimal mechanism is typically intractable due to intricate incentive compatibility constraints.

---

> > ### Author Rebuttal · Reviewer_LSUZ · 2026-04-01
> >
> > I thank the authors for their response and acknowledgement of the underlying hardness in extending their results to heterogeneous buyers.  Even as I agree that their work is a good start on this problem, it feels incomplete without some treatment of the practical setting where there are heterogeneous users.  I will stick to my original score.

---

> > > ### Author Response · Authors · 2026-04-04
> > >
> > > Thank you very much for your time and thoughtful feedback. We appreciate your recognition of our work as a foundational step. We understand your reservation, and recognize the importance of addressing practical setting with heterogeneous scaling parameters. We will expand discussion on the practical implications and limitations in the revised manuscript.

---

### Official Review · Reviewer_PG5J · 2026-03-13

**Soundness:** 4
**Presentation:** 3
**Significance:** 3
**Originality:** 3
**Overall Recommendation:** 4
**Confidence:** 4

**Summary:**

This paper studies mechanism design for selling data when buyer valuations follow scaling laws $vx^\alpha$, to capture diminishing returns to dataset size. Data is treated as a digital good: once produced, it can be sold to multiple buyers, but the seller pays a linear production cost proportional to the maximum dataset size produced. The paper analyzes three settings. In the offline setting, the authors characterize the optimal truthful mechanism. Buyers with positive ironed virtual values receive the same dataset size while others receive none. In the online setting, the authors show that simple policies (e.g., greedy or fixed production) can be arbitrarily suboptimal. They propose a two-stage mechanism that achieves a constant-factor approximation to the offline optimum. They also prove an impossibility result showing that any online mechanism can suffer a super-exponential loss relative to offline as $\alpha\to 1$. In the bilateral trade, the paper characterizes the optimal truthful mechanism and proposes simple buyer- and seller-proposing mechanisms that achieve constant-factor approximations to first-best gains from trade.

**Compliance With Llm Reviewing Policy:**

Affirmed.

**Key Questions For Authors:**

1. I would appreciate more discussion of the modeling assumptions and how slight violations of the stylized model might affect the results. In particular, it would be helpful to understand how sensitive the conclusions are to these assumptions (see also the comments in **Weaknesses**). It would also be valuable to illustrate the robustness of the results through numerical simulations.

2. The theoretical results become extremely sensitive as \(\alpha \to 1\), while empirical ML scaling laws often exhibit various forms of diminishing returns. It would be useful for the authors to discuss which empirical regimes of scaling exponents their model is intended to capture.

3. Several key results rely on nontrivial structural insights, but the paper mainly presents the formal proofs without much intuition. In several cases I only understood the key ideas after carefully reading the proofs. It would improve accessibility if the authors could provide more intuition for the impossibility result (Theorem 3.6) and the positive results in the bilateral trading section.

**Limitations:**

yes

**Strengths And Weaknesses:**

Strengths

1. The paper combines several elements rarely studied together: digital goods, endogenous data production, and scaling-law valuations. This is an interesting and timely model given current discussions around data markets for AI.

2. The paper provides clean structural characterizations in several settings. In the offline case, the optimal mechanism has an elegant structure where all served buyers receive the same dataset size. In the online setting, Theorem 3.6 shows that the gap between online and offline mechanisms can become super-exponential in $1/(1-\alpha)$, highlighting how scaling-law valuations fundamentally change the difficulty of online mechanism design.

3. In the bilateral trade section, the proposed buyer- and seller-proposing mechanisms achieve constant-factor approximations to first-best welfare. The decomposition of welfare into multiplicative terms is particularly elegant.

Weaknesses

1. The valuation model $v x^\alpha$ is clean but somewhat restrictive. It assumes a specific functional form and a common exponent across buyers, ignoring heterogeneity in scaling behavior or dataset quality. It is unclear how robust the results are to more general valuation functions.

2. The model assumes that production cost depends on the maximum allocated dataset size, which implicitly requires buyer demands to be perfectly nested (i.e., larger datasets strictly contain smaller ones). This assumption drives several results, including the pooling structure and the difficulty of the online problem. However, it may be strong in practice, since buyers may require different subsets or types of data rather than prefixes of a single dataset.

3. The paper does not include empirical or computational validation. Given the motivation from AI scaling laws, simulations or numerical examples illustrating the mechanisms under plausible distributions would strengthen the paper.

---

> ### Author Rebuttal · Authors · 2026-03-30
>
> Thank you for your constructive feedbacks.
>
> **Q1 & W1 Discussion on modeling assumptions and possibility of generalization.**
>
> Our modeling assumptions are motivated and supported by well-established scaling laws and economic models in the literature. The utility model mainly serves to isolate the key economic trade-offs arising from diminishing marginal value and private information when selling non-rival data with production costs, reflecting real-world scenarios such as data annotation for large model training. We refer to our response to Reviewer Be8v (Q1.1) for further details on the motivation and justification.
>
> We appreciate your suggestion to study the robustness of our results under violation of the assumptions or under more general valuation functions, and we acknowledge these as important directions for future work. Actually, we believe the characterization of optimal online or bilateral mechanisms (in Section 3.2, 4.2) can be extended to general concave valuation functions within the current virtual surplus maximizing framework, analogous to the offline setting. However, extending our main results concerning the approximation bounds will likely require substantially different techniques. For example, when scaling parameters are heterogeneous,  the allocation and resulting revenue of optimal mechanisms are given implicitly by an optimization problem without closed-form expressions, so the approximation targets will no longer have clear formulas. Moreover, deriving constant approximation ratios in online and bilateral settings will require significantly more involved analysis, as both Rosenthal's inequality (Lemma 3.5) and Hardy's inequality (Lemma C.2) become inapplicable. Finally, investigating sensitivity to model violations would require detailed modifications to the information assumptions (e.g. how much the seller and buyers know the violation), as well as to the assumptions about buyer behavior (e.g. definition of incentive compatibility).
>
>
> **Q2 Clarification on revenue gap when $\alpha\to 1$.**
>
> We appreciate your observation. As established in the inapproximability result (Theorem 3.6), the worst-case revenue gap between online and offline optimal mechanisms grows super-exponentially in $1/(1-\alpha)$ when $\alpha$ tends to $1$. Theoretically, this gap is inherent in information structure and inevitable when $\alpha$ is close to $1$.
>
> That said, based on practical evidence, $\alpha$ is typically small in real-world regimes such as model training, leading to moderate ratios. In a typical range of $0<\alpha<0.5$, since the optimal constant in Rosenthal's inequality is $C_p=2$ for $1<p<2$ (Line 302, right column), the approximation ratio in theorem 3.4 achieved by our proposed two-stage online algorithm is $(1+1/(1-\alpha))C_{1/(1-\alpha)}\leq 6$, which is a moderate constant.
>
> **Q3 Intuitive explaination for proofs.**
>
> Thank you for highlighting this issue. In the revised version of the paper, we will provide more intuitive explanations in the proofs to enhance readability. Specifically,
> - For theorem 3.6, to show the worst-case revenue gap between online optimal and offline optimal, we construct the "bad" instances such that each buyer's virtual value follows a $\mathrm{Bernoulli}(\frac1n)$ distribution, which offers two technical benefits. First, the binary nature of the virtual values allows an explicit, closed-form analysis for the online optimal algorithm derived via backward induction (Proposition 3.1). Second, in the limit case that $n\to\infty$, the number of buyers with positive virtual values follows a $\mathrm{Poisson}(1)$ distribution, which directly relates to the optimal constant $C_{\frac1{1-\alpha}}$ in Rosenthal's inequality.
> - For theorem 4.1 and 4.2, in addition to the welfare decomposition technique, our analysis utilizes the prefix-averaging relationship between the quantile-space representation of value (cost) distribution and virtual value (virtual cost) distribution, which allows us to apply Hardy's inequality and derive the bounds.
>
> **W2.** Thanks for your insightful comment. Our model assumes homogeneity of data, so that the valuation of a dataset only depends on its size, which captures a range of real-world scenarios such as large model training. Incorporating scaling-law valuation into heterogeneous data value models would be an interesting future direction.
>
> **W3.** Thanks for your valuable suggestion. As our primary contribution is theoretical, we derive optimal mechanisms and worst-case approximation guarantees with formal proofs, which do not rely on empirical evaluation. That said, we agree that empirical studies would provide valuable complementary insights, and we will highlight this as an important direction for future work.

---

> > ### Author Rebuttal · Reviewer_PG5J · 2026-04-03
> >
> > Thank you for the detailed response. While I acknowledge that some of my concerns have been addressed, the weakness on model assumptions and empirical evaluation is not quite convincing. I believe the model assumptions and the specific utility forms are quite essential for building the results. Meanwhile, there is no empirical evaluation which could highlight on more practical insights. I do believe this is a nice and theoretically-sound paper. I decided to maintain my score.

---

> > > ### Author Response · Authors · 2026-04-04
> > >
> > > Thank you very much for your time and careful assessment. We acknowledge your concerns regarding the model assumptions and the lack of empirical evaluation. We will thoroughly expand the discussion on the modeling choice and clarify the practical implications and limitations in the revised manuscript. We are sincerely grateful for your positive assessment of the paper's theoretical contribution.

---

### Official Review · Reviewer_5RHR · 2026-03-19

**Soundness:** 2
**Presentation:** 2
**Significance:** 2
**Originality:** 2
**Overall Recommendation:** 3
**Confidence:** 3

**Summary:**

This paper studies mechanism design for selling data as a digital good. The authors first studied an offline data selling model, and then extend to the case where one seller is selling data to incoming buyers. The results mainly include the designed incentive-compatible mechanisms. Also, the paper studies bilateral data trading under asymmetric information and shows that simple mechanisms achieve constant-factor approximations to the first-best gain from trade.

**Compliance With Llm Reviewing Policy:**

Affirmed.

**Final Justification:**

Thanks for the response. I will keep my score.

**Key Questions For Authors:**

see weakness.

**Limitations:**

yes

**Strengths And Weaknesses:**

2. Strengths
1. The paper introduces a compelling framework that captures two important properties of modern data markets: the non-rivalrous nature of data and diminishing returns implied by machine learning scaling laws. Incorporating these features leads to a meaningful departure from classical auction models.
2. The paper provides rigorous theoretical analysis across multiple settings, including offline markets, online arrivals, and bilateral trade. The results offer technically interesting contributions to mechanism design for digital goods.

3. Weaknesses
1.  The model of buyer's utility is stylish but overly simplified. The type of buyer affects the utility simply as a coefficient, while this simplifies the proof, it lacks a convincing justification. The assumption of a common and known scaling parameter $\alpha$ may be somewhat restrictive. The model assumes independent buyer types and a single scaling exponent across buyers. In practice, buyers may exhibit heterogeneous scaling behaviors depending on their models or tasks, and their valuations may also be correlated.
2. The approximation ratio of the online mechanism depends on constants derived from Rosenthal’s inequality, which grow rapidly as $\alpha$ approaches 1. As a result, the theoretical guarantees may become less meaningful in regimes where the scaling exponent is close to 1.

---

> ### Author Rebuttal · Authors · 2026-03-30
>
> We thank the reviewer for the comments.
>
> > (W1) The model of buyer's utility is stylish but overly simplified. The type of buyer affects the utility simply as a coefficient, while this simplifies the proof, it lacks a convincing justification. The assumption of a common and known scaling parameter $\alpha$ may be somewhat restrictive. The model assumes independent buyer types and a single scaling exponent across buyers. In practice, buyers may exhibit heterogeneous scaling behaviors depending on their models or tasks, and their valuations may also be correlated.
>
> Our modeling assumptions are motivated and supported by well-established scaling laws and economic models in the literature. The utility model mainly serves to isolate the key economic trade-offs arising from diminishing marginal value and private information when selling non-rival data with production costs, reflecting real-world scenarios such as data annotation for large model training. We refer to our response to Reviewer Be8v (Q1.1) for further details on the motivation and justification.
>
> Regarding the potential extensions mentioned by the reviewer: we acknowledge that they are important directions for future work, as discussed in Section 5. A direct extension to heterogeneous $\alpha$'s introduces nontrivial technical challenges; we refer the reviewer to our response to Reviewer NWYa (Q1) for a more detailed discussion. The extension to correlated valuations would require a fundamentally different mechanism design framework, which is beyond our current scope.
>
> > (W2) The approximation ratio of the online mechanism depends on constants derived from Rosenthal’s inequality, which grow rapidly as $\alpha$ approaches 1. As a result, the theoretical guarantees may become less meaningful in regimes where the scaling exponent is close to 1.
>
> We appreciate your observation. As established in the inapproximability result (Theorem 3.6), the worst-case revenue gap between online and offline optimal mechanisms grows super-exponentially in $1/(1-\alpha)$ when $\alpha$ tends to $1$. Theoretically, this gap is inherent in information structure and inevitable when $\alpha$ is close to $1$.
>
> That said, based on practical evidence, $\alpha$ is typically small in real-world regimes such as model training, leading to moderate ratios. In a typical range of $0<\alpha<0.5$, since the optimal constant in Rosenthal's inequality is $C_p=2$ for $1<p<2$ (Line 302, right column), the approximation ratio in theorem 3.4 achieved by our proposed two-stage online algorithm is $(1+1/(1-\alpha))C_{1/(1-\alpha)}\leq 6$, which is a moderate constant.

---

> > ### Author Rebuttal · Reviewer_5RHR · 2026-04-03
> >
> > I thank the authors for the response, but I am still not convinced by the justification about the utility form.

---

> > > ### Author Response · Authors · 2026-04-04
> > >
> > > Thank you very much for your time and thoughtful feedback. We understand your reservation regarding the modeling choice of utility function, and we will significantly expand the relevant discussion in the revised manuscript.

---

### Official Review · Reviewer_Be8v · 2026-03-20

**Soundness:** 3
**Presentation:** 4
**Significance:** 3
**Originality:** 3
**Overall Recommendation:** 4
**Confidence:** 4

**Summary:**

The article considers a data market setting where a seller can produce and sell data to buyers. Interestingly, data is considered as non-rivalrous, meaning that a data point, once produced, can be simultaneously sold to more than one buyer.

Buyers have private types drawn from known distributions and the seller must design a an allocation mechanism, which decides the quantity of data to allocate to each buyer, and a price mechanism, which decides the price to charge to each buyer.

As it is classical in mechanism design, the seller is restricted to use incentive compatible and individually rational mechanisms.

After defining the setting, the article studies the profit-optimal offline mechanism for the seller as if all buyers' types were declared at once, to then move to the online setting where buyers declare their types sequentially.

In the online setting, the authors characterize the optimal online algorithm, namely dynamic programming, to then propose simpler and more tractable online mechanisms for the seller. Since myopic greedy mechanisms and non-adaptative mechanisms are proved to perform poorly, as they cannot ensure constant optimality ratios with respect to dynamic programming, the authors design a two-step mechanism where first the seller produces some quantity of data depending only on the expected virtual values of the buyers, and second she greedily allocates data to each arriving agent depending on the declared value. This mechanism is proved to achieve a constant competitive ratio against the offline optimal mechanism.

Finally, the article moves to a trading setting where one seller and one buyer trade data per money, each of them having private values, and a third party decides the mechanism to be used for the data allocation and payments.

**Compliance With Llm Reviewing Policy:**

Affirmed.

**Key Questions For Authors:**

1.  Related the choice of the valuation function $V_i(v_i,x_i) = v_i x_i^{\alpha}$.

* Is this function coming from the literature ?

* How important for your results is the form of this function ? Although Thm 2.2 is generalized to more general functions, the other results are not.

2. When computing $u'_i(v_i)$ in Page 3 and in the appendix, how do you get rid of the the price $p_i$ as it also depends on $v_i$ ?

3. In Page 4, during the seller's objective subsection, why Prof considers only $D_n$ ? If the seller produces $D_1 = 5$, sells it, and then it produces $0$ during the rest of the time, $D_n = 0$ but the total produced data is $5$. Could you please explain this ?

4. At the end of Page 5 you say "...in online data markets with scaling-law valuations.", however, you have only considered one of such functions. Could you explain how you extend this conclusion ?

5. Why the two-step mechanism, in the online phase, produces additional data if this data might not be allocated to the arriving agent ? In other words, the algorithm produces data if $x_i^{\text{MG}} > D_{i-1}$ but then this data might not be allocated to buyer $i$ if her value is negative.

**Limitations:**

The article would benefit from a detailed discussion on the choice and dependence on the valuation function $V_i(v_i,x_i) = v_i x_i^{\alpha}$.

**Strengths And Weaknesses:**

**Strengths.**
- The article is well written and clearly structured.
- The article is rigorous without being unnecessarily technical.
- The considered problem is of great interest and the article is a clear fit for ICML.
- The article proposes a novel problem.
- The theoretical results of the article are sound and non trivial.

**Weaknesses.**
- The results of the article are strongly influenced by the particular choice of valuation function in Page 3, Lines 128-130, left column, which makes wonder on the robustness of the presented results.
- The article does not include any empirical validation of the theoretical results.
- The article does not discuss the limitations of its model or results.

**Minor comments**
- Page 3, Line 157, it should say ''it is required that for all $v_i, \hat{v}_i$,''
- Page 5, right column, when defining the myopic greedy mechanism, the second line in $x_i^{\text{MG}}(v_i)$ should end with a dot not a comma.
- ONL is typically denoted DP.
- OFFL is typically denoted OPT.

---

> ### Author Rebuttal · Authors · 2026-03-30
>
> Thank you for your detailed and thoughtful feedback.
>
> **Q1.1** The specific functional form we adopt is not taken directly from a single prior work, but is motivated and supported by well-established scaling laws and economic models in the literature.
>
> First, from loss scaling laws (Kaplan et al.), the test loss scales as $L(D)\propto D^{-\alpha_D}$ with $\alpha_D\approx 0.095$. If one interprets model value as inversely related to error (e.g., lower error rates enables longer correct reasoning chains), this suggests a power-law relationship between value and data size.
>
> Second, from compute-optimal scaling (Hoffmann et al.), the optimal dataset size $D$ scales with training FLOPs budget $C$ as $D\propto C^{0.5}$, and training cost is proportional to FLOPs. Combining this with a standard economic assumption (e.g., Cobb–Douglas production), where an AI company's produced economic value $V$ scales with invested expenditure $E$ as $V\propto E^\beta$, leads to a power-law relationship between value and data size of the form $V\propto D^{2\beta}$.
>
> More broadly, power-law forms are standard in economics and mechanism design (e.g., isoelastic utility and Cobb–Douglas production), further supporting our modeling choice. In addition, this structure enables tractable analysis, allowing closed-form characterizations and tight approximation guarantees.
>
> Taken together, these results provide theoretical and empirical grounding for our choice of functional form. We will clarify this connection more explicitly in the revision.
>
> [1] Kaplan et al., Scaling Laws for Neural Language Models
> [2] Hoffmann et al., Training Compute-Optimal Large Language Models Training Compute-Optimal Large Language Models
>
>
> **Q1.2** We acknowledge that our approximation results in the online and bilateral settings rely on the specific form of valuation function, while the characterization of optimal mechanisms (in Section 3.2, 4.2) can be generalized  to broader valuations, as in the offline setting.
>
> Extending these approximation results to more general forms of valuation (e.g. heterogeneous $\alpha$'s, or general concave functions) is an important direction for future work, as discussed in Section 5. However, such extensions would require substantially different technical approaches. First, when scaling parameters are heterogenous, the allocation and resulting revenue of optimal mechanisms no longer have closed forms, so the approximation targets will no longer have clear formulas. Second, deriving approximation ratios in online and bilateral settings will require significantly more involved techniques, as both Rosenthal's inequality (Lemma 3.5) and Hardy's inequality (Lemma C.2) become inapplicable. Additionally, unrestricted function forms may lead to inapproximability results in the worst case, analogous to our Theorem 3.6 when $\alpha\to 1$.
>
> **Q2**  In appendix A.1, incentive compatibility implies $U_i(v_i)\geq v_iy_i(v_i')-\tilde{p}_i(v_i')$ (Line 570). Substituting in $U_i(v_i')=v_i'y_i(v_i')-\tilde{p}_i(v_i')$ (defined at Line 566), the payment terms cancel, yielding $U_i(v_i)-U_i(v_i')\leq(v_i-v_i')y_i(v_i')$. Exchanging $v_i$ and $v_i'$, we get the inequality at Line 573, which implies $U_i'(v_i)=y_i(v_i)$ by taking $v_i'\to v_i$.
>
> **Q3**  As modeled in section 3.1, $D_i$ represents the amount of available data when serving buyer $i$, which remains available in all subsequent  rounds ($D_{i+1}\geq D_i$, Line 168 Right), since the allocation to buyer $i$ is non-exclusive. Thus, $D_n$ is exactly the total amount of produced data.
>
> **Q4** Thanks for pointing out this imprecision. We agree that our current wording ("scaling-law valuations") may be misleading, as our analysis in theorem 3.2 and 3.3 only focuses on the specific power-law form of valuation. We will revise the wording to avoid overgeneralization and clarify this point in the paper.
>
> **Q5** We apologize for a typo at Line 313, which could be misleading: $\phi_i(\hat{v}_i)\geq 0$ should be $\phi\_i^{ir}(\hat{v}\_i)\geq 0$.
> Lines 307-312 imply that any buyer $i$ with $x\_i^{MG}>D\_{i-1}$ has $r_i>0$, and thus always gets allocation $x_i=D_i$.
>
> **W2** Thank you for raising the concern about empirical validation. As our primary contribution is theoretical, we derive optimal mechanisms and worst-case approximation guarantees with formal proofs, which do not rely on empirical evaluation. That said, we agree that empirical studies would provide valuable complementary insights, and we will highlight this as an important direction for future work.
>
> **W3** Thank you for highlighting the issue, and we agree discussion of limitations could be better emphasized. While we currently discuss extensions to more general valuation functions and broader settings in Section 5, we will revise the paper to discuss these limitations and future directions more explicitly.
>
> We also appreciate your minor comments, and will revise our paper accordingly.

---

> > ### Author Rebuttal · Reviewer_Be8v · 2026-04-01
> >
> > I thank the authors for their answer. Indeed, the valuation function is an important choice for both the model and the results and it is not properly motivated during the article. I appreciate the detailed explanations during the rebuttal and I strongly suggest to add a similar discussion in the article, at least in the appendix.
> > As observed, the dependence on this function for all presented results is quite binding and unfortunately the article does not provide empirical validation in settings outside of their theoretical assumptions.
> > Still, I like the article and I believe it has several good points, so I won't downgrade my score. However, due to the risen issues, I've decided not to upgrade it either.

---

> > > ### Author Response · Authors · 2026-04-04
> > >
> > > Thank you very much for your time and positive assessment of our work. We greatly appreciate your constructive comments and will revise our paper accordingly.

---

### Decision · Program_Chairs · 2026-04-30

**Decision:**

Accept (regular)

**Comment:**

The paper studies a data market where sellers can produce and sell data to buyers. A unique feature of this market is that the seller can copy their dataset and sell it to multiple buyers. As is standard in mechanism design, the seller's mechanism must satisfy incentive-compatibility and individual rationality. The paper designs optimal mechanisms for the offline case, a constant factor approximation in the online setting, and extensions to bilateral trade.

The reviewing team appreciated the novelty and timeliness of the problem, the clean modeling, and the elegant integration of scaling laws and game theory. Thus, I recommend acceptance. We encourage the authors to include further discussion about the structure of the valuation function and buyer utility in the final version of the paper.